# Inhibition of topoisomerase 2 catalytic activity impacts the integrity of heterochromatin and repetitive DNA and leads to interlinks between clustered repeats

Michalis Amoiridis [1,2,3,4,5], John Verigos[1], Karen Meaburn[1], William H. Gittens[1], Tao Ye [2,3,4,5], Matthew J. Neale [1] & Evi Soutoglou [1] ✉

DNA replication and transcription generate DNA supercoiling, which can cause topological stress and intertwining of daughter chromatin fibers, posing challenges to the completion of DNA replication and chromosome segregation. Type II topoisomerases (Top2s) are enzymes that relieve DNA supercoiling and decatenate braided sister chromatids. How Top2 complexes deal with the topological challenges in different chromatin contexts, and whether all chromosomal contexts are subjected equally to torsional stress and require Top2 activity is unknown. Here we show that catalytic inhibition of the Top2 complex in interphase has a profound effect on the stability of heterochromatin and repetitive DNA elements. Mechanistically, we find that catalytically inactive Top2 is trapped around heterochromatin leading to DNA breaks and unresolved catenates, which necessitate the recruitment of the structure specific endonuclease, Ercc1-XPF, in an SLX4- and SUMO-dependent manner. Our data are consistent with a model in which Top2 complex resolves not only catenates between sister chromatids but also inter-chromosomal catenates between clustered repetitive elements.

The physiological processes of DNA replication or transcription impose a physical strain on the backbone of DNA, which can lead to topological stress. Unresolved topological stress can hinder faithful completion of replication and sister chromatid decatenation and therefore pose a threat to genomic integrity. Topoisomerases are enzymes that modify the topology of the DNA, and are associated with replication, transcription, and recombination[1]. Mammalian topoisomerases are divided into two classes based on their structure and catalytic mechanism: Type I, such as Topoisomerase 1 (Top1), cut one strand of the DNA, whereas Type II, such as Topoisomerase 2 (Top2), cuts both DNA strands.

Top2 enzymes introduce a double strand break (DSB) in one DNA segment and form a transient covalent bond with the free DNA overhangs. Immediately after, they pass a second DNA segment through the cut and instantly re-ligate the first segment, thus resolving issues of DNA supercoiling, knotting and catenation[2]. The two Top2 proteins, Top2α and Top2β, have divergent functions in the cell[3]. Top2α acts during replication, where it is required for proper chromosome compaction and segregation, as well as the decatenation of the sister chromatids at mitosis. Fitting with these roles, Top2α has a cell cycle dependent expression, with levels increasing in S and G2 phase, and it is overexpressed in metastatic and highly proliferating tumors[4]. On the other hand, Top2β is linked with transcription[4], and its expression remains steady throughout the cell cycle, although it is also overexpressed in tumors[5].

[1]Genome Damage and Stability Centre, School of Life Sciences, University of Sussex, Brighton BN1 9RH, UK. [2]Institut de Génétique et de Biologie Moléculaire et Cellulaire (IGBMC), Illkirch, France. [3]Institut National de la Santé et de la Recherche Médicale (INSERM), U1258 Illkirch, France. [4]Centre National de Recherche Scientifique (CNRS), UMR7104 Illkirch, France. [5]Université de Strasbourg, Illkirch, France. ✉e-mail: E.Soutoglou@sussex.ac.uk

Due to the importance of preserving genomic integrity for continuous cell proliferation, drugs which can induce topological stress have been efficiently used clinically to target cancer cells. Nevertheless, off-target side effects of these drugs highlight the critical need to better understand their mechanism of action and for elucidating how to limit their toxicity to only cancer cells. A fairly common approach in chemotherapies is the use of topoisomerase targeting drugs, of which there are two types, poisons and catalytic inhibitors. Top2 poisons stabilize the covalent bond between DNA and the Top2 protein forming DNA-Top2 cleavage complexes (Top2ccs). As a result, unrepaired DSBs are formed, causing major genomic instability, and ultimately leading to cell death[6]. In the clinical setting, Top2 poisons, such as Etoposide, are routinely used for the treatment of both solid tumors and hematologic cancers. However, even though Etoposide can efficiently target cancer cells, its use correlates with cardiotoxicity and development of secondary leukemia-associated malignancies, due to distinct chromosomal translocations[4]. On the contrary, catalytic inhibitors, such as ICRF-193, inhibit the ATPase activity of Top2, keeping it in a closed "clamp" form[4] and are proposed to only inactivate Top2, without creating Top2ccs. It remains debated whether Top2 catalytic inhibitors are capable of inducing DSBs in cells, and what effect the catalytic inactivation of Top2 might have for genomic integrity.

The genome is non-randomly arranged within the nuclear space[7]. The 3D genome organization correlates with numerous functional features such as gene activity, chromatin structure and replication timing. Although the genome wide distribution of Top2 is emerging, what genomic contexts require most Top2 activity, what parts of the genome are affected most by Top2 poisons or inhibitors, and the consequences of Top2 deregulation for human health are currently unknown. Elucidating these questions could provide vital information for the therapeutic use of Top2 drugs.

In this study, we demonstrate that catalytic inhibition of Top2 induces DNA damage at heterochromatin and repetitive DNA regions and affects the progression of the cell cycle in a Top2α-dependent manner. Moreover, whole genome sequencing reveals the genomic hotspots where the damage is induced and demonstrates that inhibition of Top2s leads to the formation of DSBs, genomic rearrangements and abnormalities associated with repetitive DNA. Our data also demonstrate that the catalytic activity of Top2α is necessary for DNA damage induction and that it is the trapping of the Top2α enzyme around chromatin that leads to the induction of the DNA damage. Most importantly, inhibition of Top2 leads to inter-chromosomal catenates of pericentric repeats from different chromosomes, which are resolved by the structure specific endonuclease Ercc1-XPF, in an SLX4- and Sumo-dependent manner.

Altogether, our results shed light into the mechanism of action of Top2 catalytic inhibitors and broaden our understanding of how they affect genomic stability and can be exploited for therapeutic purposes.

## Results

### Inhibition of Top2 catalytic activity leads to DNA damage at heterochromatin in a cell cycle-dependent manner

To study the role of Top2 activity in protecting chromatin from high torsional stress, which can lead to DNA damage, we identified the chromosomal contexts which are most affected by the catalytic inhibition of Top2. To this end, we used mouse NIH3T3 cells in which heterochromatin can be easily visualized in DAPI dense regions, known as chromocenters, and treated them with the Top2 catalytic inhibitor ICRF-193. We found that ICRF-193 mainly induces DNA damage at heterochromatin, exemplified by γH2AX strictly colocalizing with the DAPI-dense regions in 52% of treated NIH3T3 cells (Fig. 1a, b). Interestingly, inducing DNA damage at heterochromatin was specific to ICRF-193 as the Top2 poison etoposide induced DNA damage in both euchromatin and heterochromatin in 62.1% of cells, and only 11.3% of cells were found with damage enriched at heterochromatin (Fig. 1a, b).

The DNA damage induced by Top2 inhibition is not permanent, as it was repaired within 6 h after removal of ICRF-193 (Fig. 1c). The induction of damage by ICRF-193 and the ability of cells to repair this damage after the release from the drug were validated by western blot analysis (Fig. S1a). To verify that the γH2AX foci we observed colocalizing with DAPI-dense regions did indeed indicate DNA damage in heterochromatin, we visualized H3K9me3 and the heterochromatin protein HP1α. These heterochromatin markers colocalize with chromocenters in the vast majority of cells, and importantly, the percentage of cells where H3K9me3 and HP1α colocalize with the chromocenters was not significantly altered upon treatment with ICRF-193 (Fig. S1b−d). Furthermore, we performed Chromatin IP (ChIP) and analyzed the enrichment of γH2AX at major satellite repeats, which are abundant in heterochromatin. As expected, an over sixfold increase in γH2AX was observed at major satellite repeats in ICRF-193 treated cells compared to mock (DMSO) treatment (Fig. S1e).

We next sought to understand whether the induction of DNA damage depended on the stage of the cell cycle, given that DNA damage is observed only in 50% of cells. To this end, we stained NIH3T3 cells with EdU and phosphorylated H3S10 (pH3S10), as markers of S and G2/M[8] respectively (Fig. S1f) and quantified the percentage of γH2AX positive cells in each phase of the cell cycle. We found that cell cycle stage correlated with ICRF-193 induced DNA damage, as only 25% of G1 cells (EdU⁻, pH3S10⁻) showed damage, while about 50% of the S phase cells (EdU⁺ pH3S10⁻), and nearly 100% of cells in G2 (EdU⁻, pH3S10⁺), had damaged heterochromatin (Fig. S1g). Using distinct EdU patterns to differentiate between early (EdU signal covering euchromatin), mid (EdU signal co-localizing with heterochromatin), and late (EdU signal located at the periphery of the nucleus and the nucleolus) replicating cells (Fig. S1f)[9], we found that the γH2AX signal colocalized with heterochromatin in mid and late S phase suggesting that DNA damage at heterochromatin was enriched during and after replication (Fig. S1h).

For further validation of DNA damage induction based on a cell cycle stage, we used quantitative image-based cytometry (QIBC), which involves a high-content automated microscopy approach and allows the unbiased analysis of individual asynchronous cells in large cell populations (>1000 cells per condition). The cell cycle phase of each cell is classified based on DNA content and EdU incorporation. In accordance with the previous quantification by confocal microscopy, QIBC analysis for γH2AX foci revealed that ICRF-193-induced DNA damage in S and G2 phases of the cell cycle (Fig. 1d, e). Interestingly, the distribution of γH2AX perfectly correlated with the expression pattern of Top2α (Fig. S1i). These data were validated in cells synchronized in G1/S with thymidine or in G2/M with the Cdk1 inhibitor, RO-3306 (Fig. S1j, k). Despite the different spatial organization of heterochromatin in human cells, similar to our observations in mouse cells, the induction of DNA damage by ICRF-193 was most evident in S and G2 phases of the cell cycle in human bone osteosarcoma epithelia (U2OS) and human retinal pigment epithelial-1 (RPE1) cells (Fig. 1f−j). Therefore, when taken together our data demonstrate that inhibition of the catalytic activity of Top2 leads to DNA damage at heterochromatin in mid/late S and G2 phases of the cell cycle, when Top2α is expressed.

To decipher whether the induction of DNA damage requires DNA replication, EdU was supplemented into the media 15 minutes prior to ICRF-193 treatment. By adding the EdU before ICRF-193, we could ensure we analyzed the population of G2 cells in which replication had already finished before ICRF-193 was added, since we select only for 4 N cells (the EdU⁻ G2 cells (based on DAPI content)) (Fig. S1l). Interestingly, ICRF-193 induces γH2AX foci formation in 92.6% of the EdU⁻ G2 population ($p = 4.39 \times 10^{-7}$), revealing that active replication is not necessary during the treatment for DNA damage to be induced, and that the DNA damage is not necessarily passed on to G2 cells from the previous S phase (Fig. S1m, n).

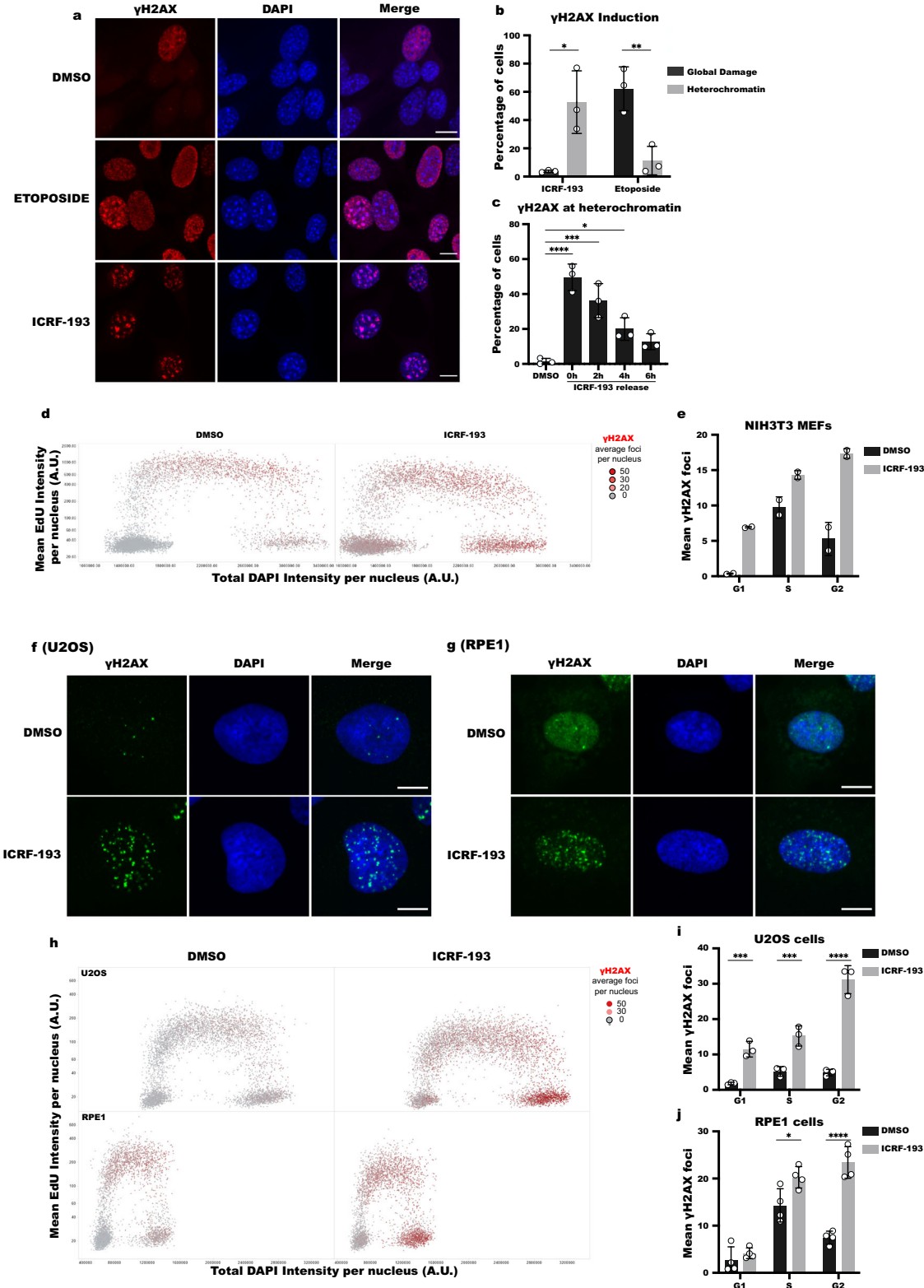

To examine how ICRF-193 affects cell cycle progression, we performed cell cycle analysis in the presence or absence of ICRF-193. As shown previously[10], we observed an increase in the percentage of G2 cells (Fig. S1o). Interestingly, we also detected a significant decrease in the number of early replicating cells, with an accompanying increase in both mid and late replicating cells (Fig. S1p). After observing the increase in replication time for heterochromatin during ICRF-193 treatment (Fig. S1o), we sought to uncover whether cells would delay the transition from S to G2 phase. To this end, we quantified the percentage of cells that are simultaneously positive for the S-phase marker EdU and the G2 marker pH3S10. There was a significant increase in the percentage of cells with this dual mark, after treatment with ICRF-193 (Fig. S1q), indicating that even if S phase is delayed, cells progress to G2 before completing replication.

Altogether, these results suggest that Top2 activity protects replicating and post-replicating heterochromatin from DNA damage, which can delay DNA replication and activate the G2-M checkpoint.

**Fig. 1 | ICRF-193 induces DNA damage at heterochromatin on a cell cycle basis. a** Confocal microscopy images of γH2AX (red) in NIH3T3 cells treated for 4 h with 5 μM ICRF-193 or 5 μM etoposide. Maximum projection of z-stacks is shown. 4′,6-Diamidino-2-Phenylindole, Dihydrochloride (DAPI; blue) was used for DNA visualization. Scale bars = 10 μm. **b** Percentage of cells with γH2AX staining localizing with heterochromatin or both with euchromatin and heterochromatin (global DNA damage). **c** Percentage of NIH3T3 cells with γH2AX at heterochromatin upon treatment with ICRF-193 and released in drug-free medium for 0, 2, 4 or 6 h. **d** Quantitative Image-Based Cytometry (QIBC) plots of γH2AX levels in NIH3T3 cells treated with DMSO or ICRF-193. Cell cycle stages were identified based on EdU and DAPI intensity. Each dot represents a single cell, and the color of the dots represents the number of γH2AX foci per cell. At least 1000 cells were analyzed per condition. A.U.: Arbitrary Units. One of the two independent biological replicates performed is shown. **e** Mean number of γH2AX foci from 2 independent biological replicates. At least 1000 cells were analyzed per condition, per replicate.

**f, g** Representative confocal microscopy images of γH2AX (green) and DAPI (blue) in **f** U2OS or **g** RPE1 cells treated with DMSO or ICRF-193. Maximum projection of z-stacks shown. Scale bars = 10 μm. **h** QIBC plots for γH2AX levels in U2OS or RPE1 cells treated with DMSO or ICRF-193. At least 2000 cells were analyzed per condition. **i, j** Quantification of the mean γH2AX foci number per nucleus in **i** U2OS and **j** RPE1 cells. For Fig. 1j, $n = 4$ biologically independent experiments. Graphs represent mean values, and error bars represent the standard deviation from at least 50 cells per condition, and 3 independent replicates unless stated otherwise. Statistical significance presented by two-sided student $t$ test in **b** ($P$ values for ICRF-193: $p = 0.0187$; Etoposide: $p = 0.009$), one-way ANOVA test in **c** (Adjusted $P$ values in comparison to DMSO for 0 h: $p < 0.0001$; 2 h: $p = 0.0003$, 4 h: $p = 0.0203$; 6 h: $p = 0.1768$), and two-way ANOVA in **i** (Adjusted $P$ values for G1: $p = 0.0007$; S: $p = 0.0005$; G2: $p < 0.0001$) and **j** (Adjusted $P$ values for G1: $p = 0.8169$; S: $p = 0.013$; G2: $p < 0.0001$): (*) $p < 0.05$, (**) $p < 0.01$, (***) $p < 0.001$, (****) $p < 0.0001$.

## ICRF-193-induced DNA damage depends on Top2α

Bisdioxopiperazines, a class of Top2 catalytic inhibitors which include ICRF-193, interact in a non-competitive way with the N-terminal site of the Top2 homodimers, and block ATP hydrolysis and enzyme turnover[11]. Since Top2α and Top2β isozymes show an extremely high degree of homology at their N-terminal domains[12], ICRF-193 could potentially effectively inhibit the function of both Top2 isozymes. However, given that γH2AX foci formation by ICRF-193 depends on the phase of the cell cycle, and that Top2α and Top2β exhibit different cell cycle expression patterns[13], we sought to investigate whether the two enzymes contribute equally to the induction of DNA damage. To this end, Top2α and Top2β were individually depleted by siRNA in NIH3T3 cells. Since Top2α is expressed from mid S-G2, and Top2β throughout the cell cycle, we depleted Top2α in G2-arrested cells and Top2β in an asynchronous population. Depletion of Top2α led to a 3.2-fold reduction of γH2AX positive ICRF-193 treated cells, when compared to the scramble (Scr) control (Fig. 2a, b). On the contrary, treatment with ICRF-193 after depletion of Top2β led to a smaller reduction of γH2AX foci (1.67-fold reduction; Fig. 2c, d). siTop2α efficiently depleted Top2α and had no effect on Top2β expression (Fig. S2a–c). Conversely, while siTop2β efficiently depleted Top2β without affecting the progression of the cell cycle, it has a small but significant off-target downregulation of Top2α levels (1.28-fold reduction; Fig. S2d–g), suggesting that the reduction in γH2AX foci in siTop2β cells, is due to the off-target reduction of Top2α. To further test the involvement of Top2β in ICRF-193 dependent DNA damage induction, we used Top2β knock-out RPE1 cells. QIBC analysis showed that γH2AX foci can still be efficiently formed in G2, even in the complete absence of Top2β, albeit to a slightly reduced level (Fig. 2e, f). The relevance of Top2β was further validated in Top2β knock-out MEFs, where we observe that γH2AX foci can still be efficiently formed upon ICRF-193 treatment, even in the complete absence of Top2β (Fig. S2h–j). Finally, the involvement of Top2α in ICRF-193-induced DNA damage was further confirmed in HCT116 OsTIR1[14] expressing cells, in which endogenous Top2α was fused to an auxin inducible degron tag (Top2αAID) to induce its rapid depletion. After ensuring that Top2α is efficiently depleted upon 1 h treatment with Indole-3-acetic acid (IAA) (Fig. S2k), the induction of DNA damage in these cells was verified by immunofluorescence and QIBC analysis of the mean number of γH2AX or 53BP1 foci and corroborated that Top2α depletion prevents the induction of ICRF-193-induced DNA damage (Fig. 2g–i). Altogether, these results suggest that the ICRF-193-induced DNA damage primarily depends on Top2α, and Top2β has a minor role in the induction of DNA damage.

Next, after demonstrating that the targeting of Top2α by ICRF-193 is necessary for the induction of DNA damage, we explored whether the over-expression of Top2α is sufficient to trigger DNA damage in the presence of ICRF-193. To this end, we assessed if treatment with ICRF-193 induces DNA damage in G1 NIH3T3 cells, where the Top2α and DNA damage levels in heterochromatin are physiologically low.

Interestingly, ectopic overexpression of Top2α-YFP, led to a very small increase (7.7%) in G1 cells which showed DNA damage at heterochromatin (Fig. S2l).

When taken together, these results suggest that ICRF-193-induced DNA damage depends on Top2α, but its presence alone is not sufficient, and other factors may contribute to the induction of DNA damage, such as collision of the trapped Top2 proteins with the replication machinery and/or inhibition of sister chromatid decatenation[15].

## ICRF-193 induces DSBs at repetitive sequences

Top2 enzymes bind to the genome regardless of chromatin status[16–19]. The main cause of DNA damage induction by Top2 poisons in G1 cells is the collision of Top2ccs with the transcription machineries in euchromatin[20,21]. Our results also show that the trapping of the Top2 enzymes by etoposide can induce damage anywhere in the nucleus (Fig. 1a). Therefore, it is unexpected that treatment with ICRF-193 shows such a concentrated response in the heterochromatic domains of mouse cells (Fig. 1a). To explore the possibility that due to the low sensitivity of immunofluorescence we may have failed to detect γH2AX enrichment in non-chromocenter genomic regions of the genome after ICRF-193 treatment, we performed γH2AX ChIP-seq in NIH3T3 MEFs treated with ICRF-193 or DMSO. To validate our approach, we first visualized the pericentric region of chromosome 9, which locates to chromocenters and thus γH2AX enrichment is expected (Fig. 3a). Indeed, broad γH2AX peaks spanning megabases were detected in this region in cells treated with ICRF-193 compared to DMSO (Fig. 3a).

Next, we examined whether ICRF-193 can induce DNA damage only in major satellite repeats, which are abundant in pericentric regions, or if it has a broader effect on repetitive sequences. DESeq2 analysis on data from two biological replicates showed that ICRF-193 led to a 5.3-fold enrichment of reads in major satellite repeats, compared to the DMSO control (Fig. 3b). This observation is in accordance with our previous results and verifies the robust DNA damage induction we detect at heterochromatin by microscopy. We then assessed other tandem repeats containing nuclear compartments, such as centromeric minor satellite repeats, or small subunit (SSU) and large subunit (LSU) rDNA, which are found in the nucleolus. Interestingly, we also detected a significant increase in the number of reads for these regions after treatment with ICRF-193 (Fig. 3c–e). Additionally, we observed an enrichment in the number of reads for several other classes of repetitive sequences, such as Long Terminal Repeats (LTRs) (Table S1). Interestingly, our ChIP-seq analysis also validates that γH2AX levels are reduced after ICRF-193 is removed from the culturing media (Fig. 3b–e), showing that cells can repair the induced damage at heterochromatin.

To gain a fuller picture of the DNA damage hotspots and to identify if there is any preference for euchromatin or heterochromatin, we plotted the number of γH2AX peaks according to their position in the genome. In promoters, introns, and exons, we did not detect a

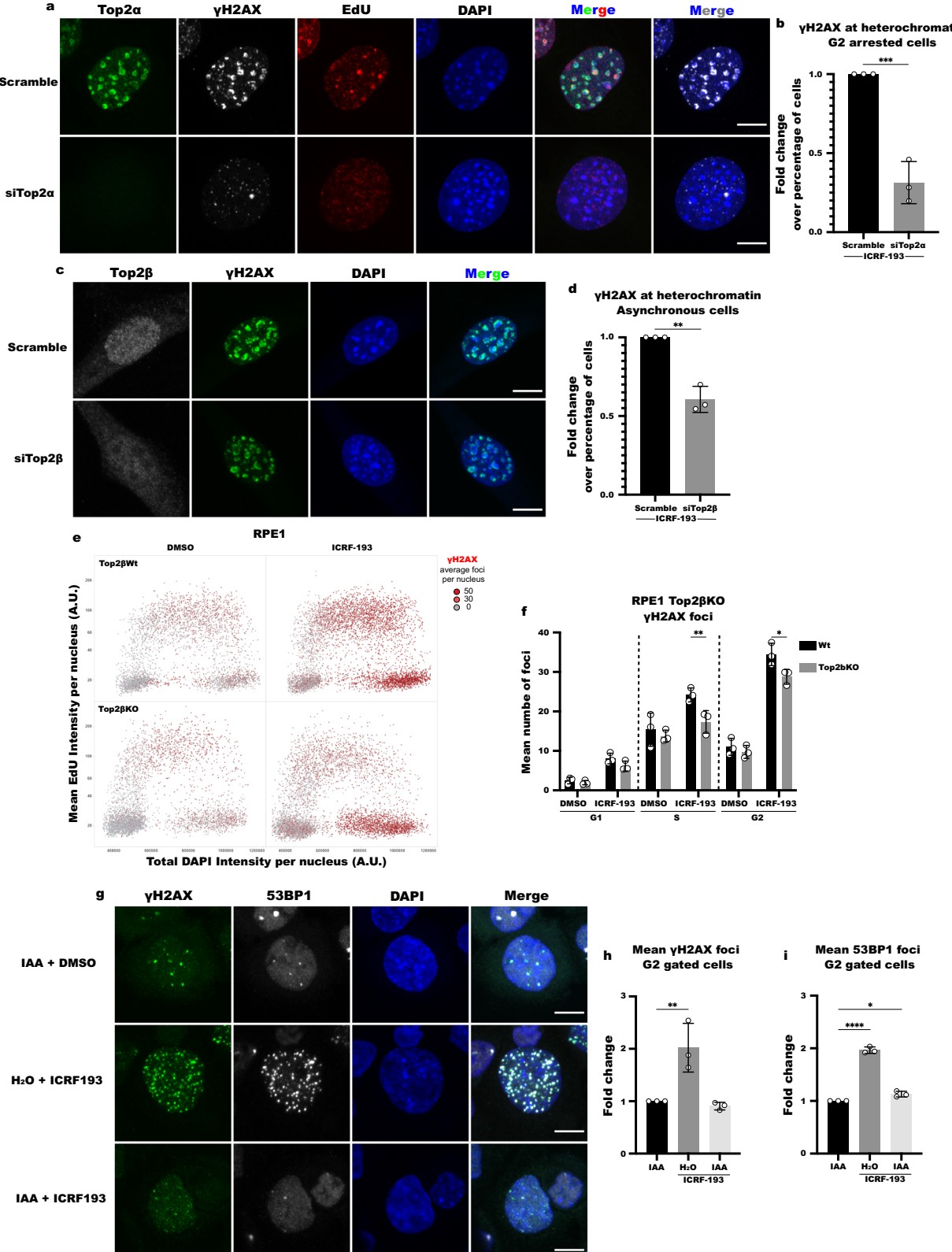

consistent increase in the number of peaks after treatment with ICRF-193 in either biological replicate (Fig. 3f, g), thus providing no evidence for targeted DNA damage induction in euchromatin. Even though the mapping of peaks to repetitive sequences is not entirely accurate, since the reads are randomly mapped in each region with the same sequence, we detected a consistent increase of γH2AX peaks in heterochromatic regions, such as SINEs, LINEs and LTRs (Fig. 3f, g).

To further validate these observations, we segmented the genome into 10 kb bins and, based on the sequence content in the center of the bin, classified them as repetitive, intergenic or active sites (TSS, TTS, Exons, Introns; see Material and Methods). We subsequently performed a differential coverage analysis using DESeq2 (1.38.3) to quantify the number of bins that showed an over 1.5 $\log_2$ fold enrichment (cutoff is for ICRF-193 0 h release over DMSO: $\log2fc > 1.5$ or

**Fig. 2 | ICRF-193 induces damage by mainly targeting Top2α. a** Representative confocal microscopy images of Top2α (green), γH2AX (white), EdU (red) and DAPI (blue) in G2- arrested NIH3T3 cells transfected with scramble siRNA or siTop2α and treated with ICRF-193. **b** Quantification of mid/late S or G2 phase NIH3T3 cells with γH2AX at heterochromatin. **c** Representative confocal microscopy images of Top2β (white), γH2AX (green) and DAPI (blue) in NIH3T3 cells transfected with scramble siRNA or siTop2β and treated with ICRF-193. **d** Quantification of NIH3T3 cells with γH2AX at heterochromatin at the indicated conditions. **e** QIBC plots of γH2AX foci number in Top2β wt and knock-out (KO) RPE1 cells treated with DMSO or ICRF-193. At least 1000 cells were analyzed per condition, per replicate. **f** Quantification of mean numbers of γH2AX foci per cell. **g** Representative confocal microscopy images of γH2AX (green), 53BP1 (white) and DAPI (blue) in HCT116-OsTIR1/ Top2αAID cells. **h** Quantification of the fold change of mean numbers of γH2AX foci in comparison to the control (IAA), for G2 gated cells based on their DAPI intensity levels. At least 800 cells were analyzed per condition, per replicate. **i** Quantification

of the fold change of mean numbers of 53BP1 foci in comparison to the control (IAA), for G2 gated cells based on their DAPI intensity levels. At least 800 cells were analyzed per condition, per replicate. Graphs represent average values, and the error bars represent the standard deviation from at least 50 cells per condition and 3 independent replicates unless stated otherwise; for the QIBC analysis at least 1000 cells were used per condition and 3 independent replicates unless stated otherwise. For confocal images maximum projection of z-stacks is shown; scale bars = 10 μm. Statistical significance presented by two-sided student $t$ test in **b** (P value = 0.00087) and **d** ($P$ value = 0.0012), two-way ANOVA test in **f** (Adjusted $P$ values for DMSO G1: $p = 0.9998$; ICRF-193 G1: $p = 0.8682$; DMSO S: $p = 0.9307$; ICRF-193 S: $p = 0.0037$; DMSO G2: $p = 0.9767$; ICRF-193 G2: $p = 0.0201$), and one-way ANOVA test in **h** (Adjusted $P$ values in comparison to DMSO for $H_2O$ + ICRF-193: $p = 0.0066$; IAA + ICRF-193: $p = 0.8832$) and **i** (Adjusted $P$ values in comparison to DMSO for $H_2O$ + ICRF-193: $p < 0.0001$; IAA + ICRF-193: $p = 0.0271$): (*) $p < 0.05$, (**) $p < 0.01$, (***) $p < 0.001$, (****) $p < 0.0001$.

---

<−1.5, adjusted p value < 0.05). We observed 128 bins with a significant enrichment of γH2AX (Fig. S3a). The majority of these bins were annotated to repetitive sites, with only a very small portion of them correlating to active sites (Fig. S3b, c). It is noteworthy that not every type of repetitive DNA site is affected after ICRF-193 treatment. For example, we did not observe an increase of γH2AX reads in all classes of LTRs, such as ORR1C1; in transposable elements, such as L1MDA, or L1M4c; or in simpler types of repeats, such as (ATG)n or (CCTG)n (Table S1).

Altogether, these results show that the vast majority of the DNA damage caused by ICRF-193 is induced in the repetitive genome, with only a very small portion of the active genome affected.

While γH2AX is a strong indicator of the DNA damage response (DDR) after the formation of DSBs, it is also tightly linked to the responses to replication stress[22–24]. To determine conclusively whether ICRF-193 treatment leads to induction of DSBs, we first examined the recruitment of DSB repair factors 53BP1, BRCA1 and RPA32 at damaged heterochromatin (Fig. S3d–g). Consistent with γH2AX, 53BP1 and BRCA1 were enriched at heterochromatin after ICRF-193 treatment (26% and 28.2% of cells respectively), with the percentage of cells with colocalization decreasing after release from the drug. Interestingly, there was a small but significant increase in the recruitment of the ssDNA binding protein RPA32 2 h after drug release (Fig. S3d, g). Similar results for 53BP1 foci were obtained in human U2OS and RPE1 cells (Fig. S3h–j). Comparable with the γH2AX foci pattern, 53BP1 foci are induced in a cell cycle-dependent manner, with the highest signal increase found in G2 (4.3-fold for U2OS & 3.27-fold for RPE1; Fig. S3i, j). These results suggest DSBs are indeed being formed.

To directly prove DSB formation we performed neutral comet assay. Consistent with DSB induction, treatment with ICRF-193 led to a significant increase of the tail moment, showing that DSBs are formed and persist 4 h after the removal of the drug (Fig. 3h). Even though the comet assay allows us to detect an enrichment of DSBs, it fails to provide information regarding their nuclear position. For this reason, we supplemented our results with the imaging technique called exo-FISH[25], which allows us to pinpoint the spatial position of DSBs. Exo-FISH utilizing a fluorescent probe that anneals at major satellite repeats, showed that the majority of breaks induced by ICRF-193 located at regions of heterochromatin (Fig. 3i, j), which is consistent with our γH2AX ChIP-seq and imaging data.

We next aimed to decipher the major kinases involved in the DDR after ICFR-193 treatment. We found that for up to two hours after the release of the drug the induction of γH2AX primarily relies on ATM (Fig. S3k). Subsequently, ATR is important for the phosphorylation of histone H2AX, until the repair is finished (Fig. S3l). These two kinases seem to be the only enzymes responsible for the initiation of DDR, since combinational treatment of ATM inhibitor and ATR inhibitor completely blocked the induction of γH2AX by ICRF-193 (Fig. S3m).

To understand which pathways are involved in repairing ICRF-193-induced DNA damage, we followed the γH2AX kinetics upon drug release in Ku80 knock-out (Ku80KO) MEFs, where the non-homologous end joining (NHEJ) pathway is completely abolished. We found that these cells are incapable of repairing ICRF-193 dependent damage, even after 6 hours of drug release (Fig. S3n). The importance of NHEJ was further validated by inhibition or transient depletion of DNAPK, which showed a significant delay in the repair after 4 or 6 hours of drug release (Fig. S3o–q).

To investigate the contribution of recombination mechanisms in the repair, we targeted RAD51 and RAD52 recombinases through depletion or inhibition[26] (Fig. S3p, r–t). Notably, while depletion or inhibition of RAD52 did not alter γH2AX resolution upon ICRF-193 treatment (Fig. S3p, s), depletion of RAD51, albeit not inhibition, led to a delay in γH2AX resolution (Fig. S3p, t). These findings suggest a minor role for recombination mechanisms in repairing DNA damage induced by Top2 inhibition.

These results demonstrate that ICRF-193 leads to the induction of DSBs in heterochromatin, which are primarily repaired by NHEJ.

## ICRF-193-induced DNA damage in interphase jeopardizes genome integrity

DNA damage can be lethal for cells since it can severely affect genomic stability. DSBs are the most toxic type of lesions, which can enhance the chances of occurring translocations and impair proper chromosome segregation. Therefore, we next sought to investigate the consequences of the ICRF-193-induced DNA damage observed in heterochromatin during interphase on genomic instability. We performed fluorescence in situ hybridization (FISH) to visualize major satellite repeats in metaphase spreads from NIH3T3 cells treated with ICRF-193 while in interphase and in which the drug had been removed to allow progression to mitosis (Fig. S4). We found several types of chromosomal abnormalities associated with hetero-chromatin malformations (Fig. 4a). These include: i) chromocenter malformations where the pericentromeric regions have lost their round shape, or DNA breaks where the whole pericentromeric region was detached from the rest of the chromosome arm; ii) whole chromosome translocations, where two telocentric mouse chromo-somes are conjugated at their centromeric sites giving rise to a single metacentric chromosome; iii) heterochromatin aggregates, where single indistinct masses formed from the chromocenters of different chromosomes; iv) heterochromatin DNA bridges forming between two neighboring chromocenters. The last two types of abnormalities are indicative of failed decatenation by Top2α at the pericentric region. Quantification of the metaphases revealed a significant increase for all the pericentric malformations by ICRF-193 (fold increase over DMSO: 7.4 for malformations, 2.9 for Translocations, 4.3 for aggregates, 3.7 for bridges; Fig. 4b).

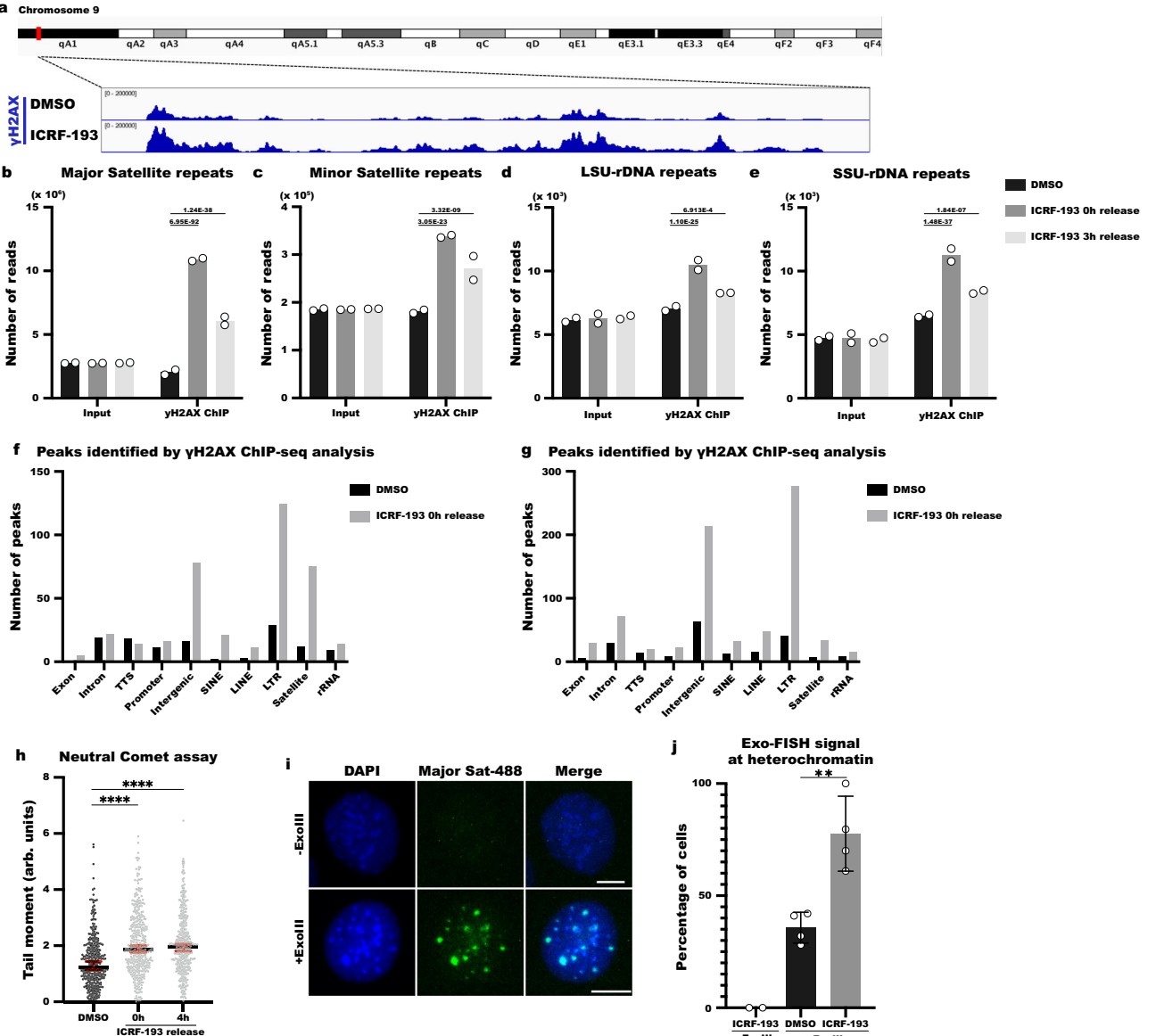

**Fig. 3 | ICRF-193 induces DSBs in repetitive sequences. a** Representative image of the pericentric region from chromosome 9, mouse mm10 genomic assembly, showing the increase in number of reads with γH2AX signal after treatment of NIH3T3 cells with ICRF-193. **b–d** Quantification of the increase in number of reads after γH2AX-ChIPseq analysis in NIH3T3 cells mapped to Major (**b**) and Minor (**c**) Satellite repeats, or to Long (**d**) and Small (**e**) subunits of ribosomal RNA repeats. Graphs represent average values from n = 2 biologically independent experiments. Statistical significance presented with adjusted p values by comparison with two-sided DE-Seq-2 analysis. **f, g** Quantification of the number of peaks identified for the indicated genomic sequences in NIH3T3 cells treated with DMSO (black) or ICRF-193 (gray) in the first (**f**) and second (**g**) biological replicates. Peak calling was performed by MACS2 and they were annotated by HOMER to general categories of genomic regions such as Exons, Introns, Promoters and transcription termination sites (TTS) and repetitive regions such as SINEs, LINEs, LTRs, Satellite and rRNA repeats. **h** Quantifications of tail moment in neutral comet assays performed in

NIH3T3 cells. 4 biological replicates were performed for each condition, with measurements taken from 100 cells per replicate, per condition. Statistical significance presented by one-way ANOVA test. (Adjusted *P* values in comparison to DMSO for 0 h: *p* < 0.0001; 4 h: *p* < 0.0001). Bars represent the median value, and the error bars represent 95% CI. **i** Representative confocal microscopy images of exo-FISH signal (green), and DAPI (blue), with (+) or without (−) ExonucleaseIII (ExoIII) processing in NIH3T3 cells. Maximum projection of z-stacks is shown. Scale bars = 10 μm. **j** Percentage of NIH3T3 cells with exo-FISH signal colocalizing at heterochromatin, as shown in Fig. 3i. 2 biological replicates were performed for −ExoIII condition and 4 biological replicates for +ExoIII conditions. At least 20 cells were counted per replicate in −ExoIII and at least 50 cells were counted per replicate for the +ExoIII condition. The graph bars represent average values, and the error bars in the graphs represent the standard deviation. Statistical significance presented by two-sided student *t* test (*P* value for +ExoIII treatments: *p* = 0.0036). (*) *p* < 0.05, (**) *p* < 0.01, (***) *p* < 0.001, (****) *p* < 0.0001.

To investigate whether the cells carry these genomic aberrations to the next cell cycle, we quantified the percentage of cells with micronuclei that contain pieces of heterochromatin, and/or with DNA bridges that still connect two daughter nuclei in the next G1 phase (Fig. 4c). We observed a 6.2-fold increase in the total number of micronuclei (*p* < 0.0001), and a 13.9-fold increase in micronuclei containing major satellite repeat heterochromatin (Fig. 4d). Furthermore,

even though it is a rare event to find DAPI DNA bridges in the control conditions, a large increase was observed after releasing ICRF-193 (*p* < 0.01), and the majority (69%) were positive for major satellite repeat heterochromatin (Fig. 4e).

Altogether, these results reveal that ICRF-193 induces damage in heterochromatin during interphase, which can lead to a significant level of genomic instability that is passed on to daughter cells.

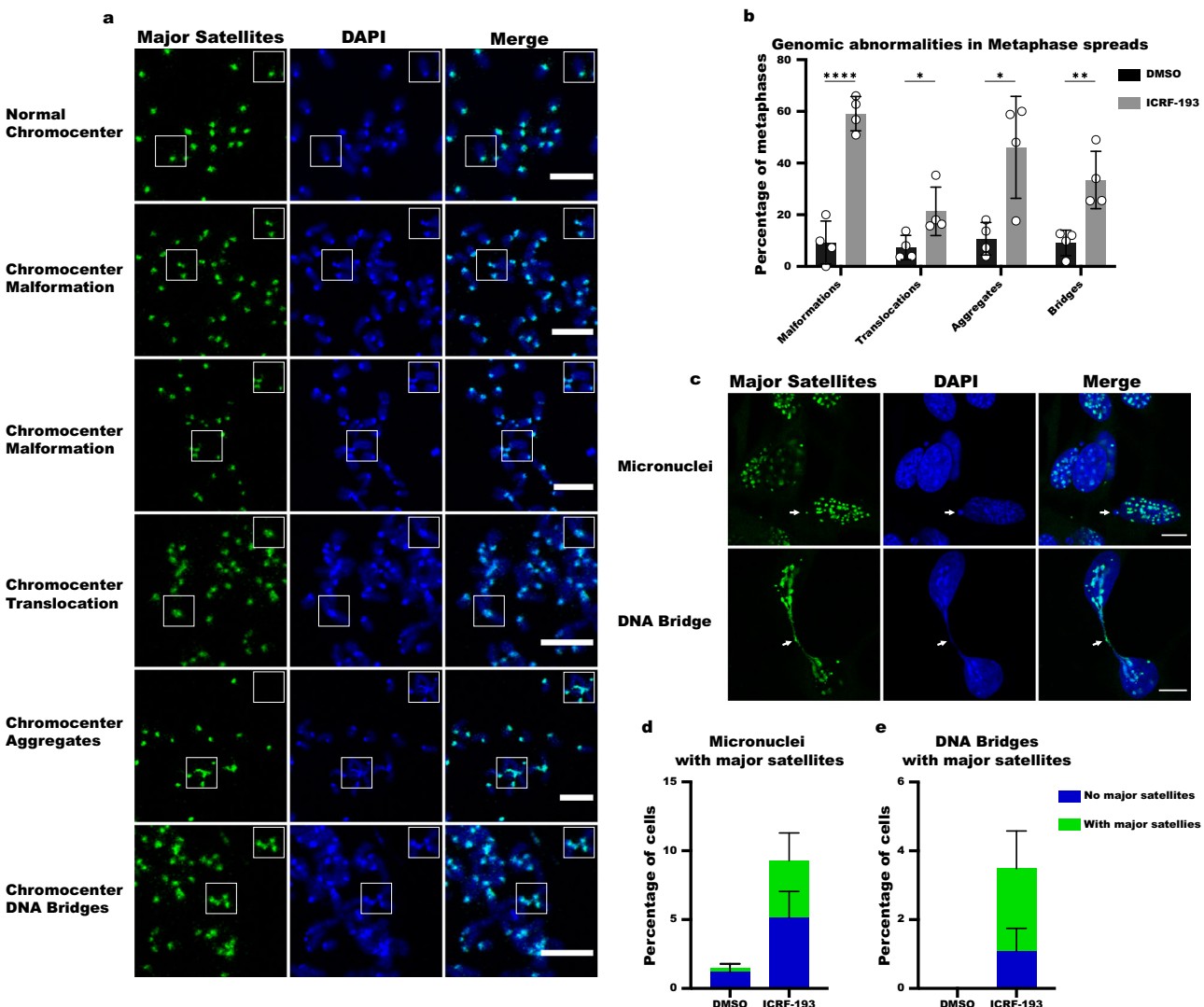

**Fig. 4 | ICRF-193 induces genomic instability at heterochromatin.**
**a** Representative confocal microscopy images of metaphase chromosome spreads from NIH3T3 cells. FISH was utilized to visualize Major Satellite repeats (green). DNA is visualized with DAPI (blue); examples of the different classes of genomic abnormalities detected are shown. Scale bars = 10 μm. **b** Quantification of the abnormalities induced by DMSO or ICRF-193; n = 4 biologically independent experiments. **c** Representative confocal microscopy images of micronuclei and DNA bridges formed in NIH3T3 cells treated with DMSO or ICRF-193; Major Satellite repeats (green), and DNA (blue). Scale bars = 10 μm. **d** Quantification of the percentage of cells with micronuclei containing Major Satellites DNA (green bar) or not

(blue bar). At least 500 cells were counted per condition. n = 3 biologically independent experiments. **e** Quantification of the percentage of cells with DNA bridges that contain Major Satellites DNA (green bar) or not (blue bar). At least 500 cells were counted per condition. n = 3 biologically independent experiments. All graph bars represent average values, and the error bars in the graphs represent the standard deviation from at least 50 metaphases per condition, and 3 independent biological replicates unless stated otherwise. Statistical significance presented by two-sided student *t* test in **b** ($P$ values for Malformations: $p < 0.0001$; Translocations: $p = 0.0367$; Aggregates: $p = 0.0142$; Bridges: $p = 0.007$): (*) $p < 0.05$, (**) $p < 0.01$, (***) $p < 0.001$, (****) $p < 0.0001$.

## ICRF-193 has a differential effect in the dynamics of the Top2s in chromatin

To gain further insights into the mechanism by which Top2 inhibition leads to DNA damage, we investigated the impact of ICRF-193 on Top2α and Top2β localization and dynamics on heterochromatin. In accordance with previous reports[17,18], we found that both Top2α and Top2β are physiologically enriched at sites of heterochromatin (Fig. 5a). Interestingly, after treatment with ICRF-193 we observed that Top2α stays bound on DNA, and the number of cells with Top2α at heterochromatin is increased (Fig. 5a, b). On the contrary, the percentage of cells with Top2β is reduced with ~50% of cells having lost the expression of Top2β after ICRF-193 treatment, compared to the control (Fig. 5a, c).

To further corroborate these findings, we performed chromatin fractionation assays. The efficiency of these assays in separating the

various protein fractions was verified based on the levels of lamin A or histones H3 and H4 found on chromatin, and GAPDH or Tubulin found in the soluble fractions (Fig. 5d, e). When the fractionation was performed with a low salt containing buffer, we observed that ICRF-193 leads to an increase of Top2α on chromatin, and a downregulation of Top2β in both the nucleoplasm and on chromatin, as detected by immunofluorescence (Fig. 5a, d). To assess to what extent ICRF-193 can trap the Top2 enzymes on the chromatin, we performed the fractionation assays with high salt containing buffer, in which all proteins that are not covalently linked or trapped on the DNA are removed and partition to the soluble fraction. Top2α and Top2β were found in the soluble fraction, in untreated conditions however, upon treatment with ICRF-193, the vast majority of Top2α stayed bound to chromatin (Fig. 5e). This is also the case for the remaining Top2β fraction, which does not get degraded and remains bound to chromatin in the

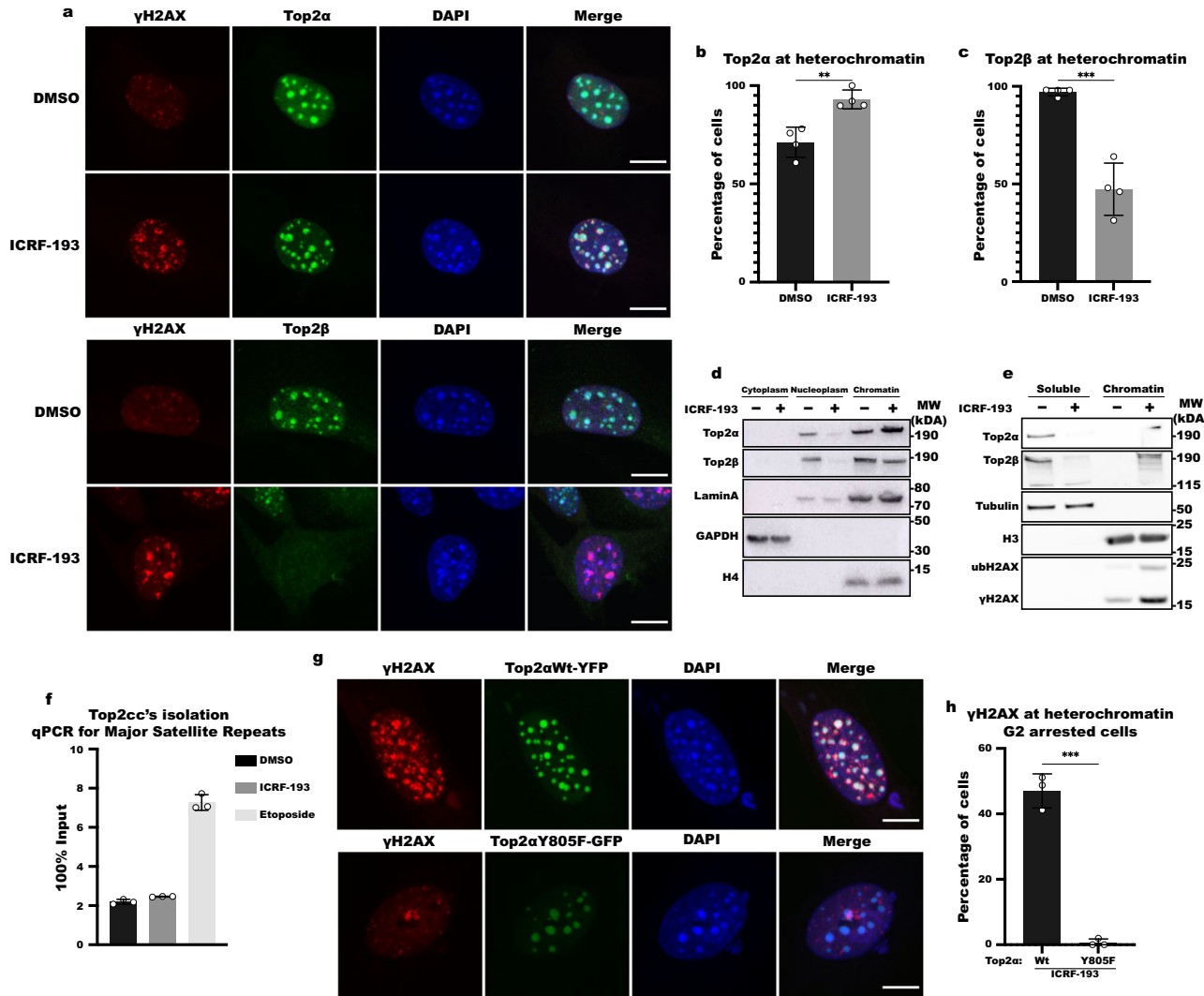

**Fig. 5 | ICRF-193 affects the localization of Topoisomerase 2 α & β.**
**a** Representative confocal microscopy images of γH2AX (red), Top2α (upper panel; green), Top2β (lower panel; green) and DAPI (blue) in NIH3T3 cells, treated with DMSO or ICRF-193. Maximum projection of z-stacks is shown. Scale bars = 10 μm. **b**, **c** Quantification of NIH3T3 cells with Top2α (**b**) or Top2β (**c**) at heterochromatin; n = 4 biologically independent experiments. **d**, **e** Western blot analysis of Top2α and Top2β in chromatin fractionations from NIH3T3 cells treated with DMSO or ICRF-193, with 200 mM (**d**) or 500 mM (**e**) NaCl containing buffer. **f** Quantification of Top2-cc's at Major Satellite repeats in NIH3T3 cells treated with DMSO and ICRF-193 or etoposide. **g** Representative confocal microscopy images of γH2AX (red), Top2αWt-YFP (top; green), Top2αY805F-GFP (bottom; green) and DAPI (blue) in

G2-arrested NIH3T3 cells, transfected with Top2αWt or Top2αY805F and treated with ICRF-193. Maximum projection of z-stacks is shown. Scale bars = 10 μm. **h** Quantification of the percentage of Top2αWt or Top2αY805F G2-arrested cells, after ICRF-193 treatment, with γH2AX at heterochromatin. For Western blots Lamin A, GAPDH, Tubulin, H3 & H4 were used as fraction specific loading controls. All graphs represent average values, and the error bars in the graphs represent the standard deviation from at least 50 cells per condition, and 3 independent biological replicates unless stated otherwise. Statistical significance presented by two-sided student t test in **b** ($p = 0.003$), **c** ($p = 0.0003$), and **h** ($p = 0.0001$): (*) $p < 0.05$, (**) $p < 0.01$, (***) $p < 0.001$, (****) $p < 0.0001$.

presence of ICRF-193 (Fig. 5e). The induction of DNA damage was verified by γH2AX and ubiquitinated H2AX (Fig. 5e).

While Top2 poisons trap the Top2 enzymes when they are covalently linked to the DNA, Top2 catalytic inhibitors, such as ICRF-193, are reported to trap them in the closed clamp form around chromatin[27]. Since we find that ICRF-193 keeps the Top2 enzymes tightly linked to chromatin, even in high-salt conditions, we sought to test whether ICRF-193 induces Top2ccs using CC-seq[16]. This technique is powerful because it can detect even transient Top2ccs, which are not stabilized by chemical substances. Indeed, while treatment with etoposide led to the expected increase of Top2cc signal at major satellite repeats (NIH3T3 cells), satellite II, satellite III and centromeric repeats (U2OS cells), no increase was observed upon ICRF-193 treatment (Fig. 5f, S5a, b). Therefore, we conclude that ICRF-193 traps Top2 on heterochromatin in the closed clamp form, without inducing Top2ccs.

To further support these findings, we performed FRAP analysis to determine how ICRF-193 affects the dynamics of Top2a in heterochromatin in the presence of ICRF-193. Indeed, we observed that ICRF-193 drastically decreases the mobility of Top2α fused to YFP, suggesting that catalytic inhibition of Top2α leads to its avid binding to chromatin (Supplementary Movies 1 and 2; Fig. S5c).

To establish if the catalytic activity of Top2 is important for the induction of DNA damage by ICRF-193, we followed the formation of γH2AX foci at heterochromatin in G2-arrested NIH3T3 cells treated with ICRF-193, expressing either Top2α Wt or Top2αY805F, a dominant negative mutant which lacks the catalytic activity[28]. Interestingly, unlike control cells over-expressing Top2αWt, we failed to detect γH2AX foci at heterochromatin in cells over-expressing the mutated form of the enzyme (Fig. 5g, h), suggesting that the catalytic activity of Top2α is necessary for the induction of damage by ICRF-193.

These results demonstrate that ICRF-193 leads to the prompt degradation of Top2β and the trapping of Top2α on heterochromatin, and that the catalytic activity of Top2α is necessary for DNA damage induction. Our results once again reveal the differential effect ICRF-193 has on the two Top2 isozymes.

## Sumoylation has an important role in the cellular response to treatment with ICRF-193

Sumoylation was shown to drive the removal of DNA-protein cross-links (DPCs)[29–31]. By regulating the ubiquitination of proteins covalently trapped on DNA, it can catalyze their proteasome-dependent degradation[29,31]. This was also shown to be the case for Top2ccs stabilized by etoposide, where the trapped Top2 enzymes are SUMOylated and subsequently ubiquitinated in order to be removed by proteasome-dependent degradation from the DNA[32]. In addition, inhibition of sumoylation or ubiquitination was reported to reduce the induction of γH2AX by etoposide[31,33]. Whether sumoylation plays a role in the removal of trapped Top2 from the DNA or in the induction of DDR and repair by ICRF-193 remains unknown.

To elucidate if sumoylation affects the induction of γH2AX, we treated NIH3T3 cells with the sumoylation inhibitor, ML-792 (Sumoi), or with DMSO as a control, prior to the addition of ICRF-193. We observed a significant reduction of γH2AX foci for all phases of the cell cycle (Fig. 6a). Noticeably, the highest effect was found in G2 phase (15.8-fold reduction), where we also observed the highest induction of DNA damage by ICRF-193 (Figs. 1e, i, j, S1d, h, 2f, S2j). For this reason, we also tested Sumoi in NIH3T3 cells synchronized in G2 and confirmed the reduction of γH2AX foci (8.5-fold decrease; Fig. S6a, b). To further validate this observation, we overexpressed Sumo1, Sumo2 and Sumo3 proteins, fused to RFP, in G2 arrested NIH3T3 cells. We observed an ICRF-193-dependent recruitment at heterochromatin in 5.1% of cells for Sumo1, 32% for Sumo2 and 36.6% for Sumo3. The recruitment was abolished in Sumoi treated cells (Fig. S6c, d). Therefore, our data suggest sumoylation has an important role in the induction of γH2AX by ICRF-193. The effect on γH2AX was not due to changes in the levels of Top2α (Fig. S6e, f). Furthermore, we confirmed that both Top2 isozymes remain trapped on chromatin by fractionation assay, with a slight increase in the levels of Top2α on chromatin upon pre-treatment with Sumoi (Fig. 6b). Finally, FRAP analysis did not show any difference in the mobility of Top2α with or without the presence of Sumoi, thus further supporting our findings (Supplementary Movie 3; Fig. S6g). Altogether, our results show that sumoylation has an important role in the induction of DNA damage response by ICRF-193, without affecting the localization of the Top2 enzymes.

Recently, sumoylation was shown to have a role in the degradation of Top2ccs, stabilized by topoisomerase poisons[31]. In agreement with this, we observed by both Western blot and microscopy analysis, that Sumoi prevents the degradation of Top2β in ICRF-193 treated NIH3T3 MEFs (Figs. 6b and S6h, i). Hence, to test whether Sumoi prevents the ICRF-193 dependent γH2AX formation by inhibiting the degradation of trapped Top2s by the proteasome, and thus not revealing DSBs left behind by the cleaved DNA protein crosslinks, we used the proteasome inhibitor MG132. The proteasome inhibition led to a slight but significant increase of Top2α foci and an evident increase in the number of Top2β foci in G2-arrested NIH3T3 MEFs (Fig. S6j, k). Furthermore, the impact of proteasome inhibition led to a significant decrease of γH2AX foci (1.5-fold) induced by ICRF-193 (Fig. S6l), however it was not as potent as that of Sumoi (Fig. S6b).

Since there was a significant rescue of the Top2β levels, we wanted to explore further whether the inhibition of ICRF-193-dependent Top2β degradation by Sumoi is responsible for the reduction of γH2AX foci. To this end, we pre-treated both Top2β Wt and KO RPE1 cells with the Sumoi, and subsequently treated them with ICRF-193 to compare how stabilization of Top2β would affect the induction of γH2AX. Treatment with Sumoi reduced the γH2AX induction in a similar way,

both in the presence (2.9-fold decrease in G2 cells) or absence (2.7-fold decrease in G2 cells) of Top2β (Figs. 6c, d, e, S6m). Hence, we conclude that Top2β degradation by ICRF-193 does not contribute to the γH2AX formation initiated by sumoylation signaling.

So far, we have shown that Sumoi reduces the γH2AX levels induced by ICRF-193, without reducing the trapping of Top2α around chromatin. Thus, sumoylation either reduces the levels of DNA damage or has an impact on the response and signaling to DNA damage, which might augment genomic instability. To investigate which scenario holds true, we performed metaphase spreads to check the genomic stability induced by ICRF-193, in the presence or absence of Sumoi. If Sumoi affects the induction of DNA damage, the combination of the two drugs should decrease the aberrations we found on metaphase chromosomes. However, if sumoylation plays a role in the DDR induced by ICRF-193, pre-treatment with Sumoi should increase the genomic instability even further, since the cells will carry severe damage and normally continue their cell cycle, as they will have no means to detect it. We found that pre-treatment with Sumoi led to an increase in both chromosomal translocations and chromocenter bridges induced by ICRF-193 (Fig. 6f). These observations suggest that Sumoylation dependent mechanisms protect against genomic instability caused by Top2 inhibition.

## Ercc1-XPF endonuclease safeguards genomic stability by resolving inter-chromosomal DNA bridges

To understand the mechanism by which sumoylation protects against ICRF-193-induced genomic instability, we searched for DNA repair factors that are regulated at sites of damage in a SUMO-dependent manner. Ercc1 works in a complex with XPF to form a structure-specific endonuclease that takes part in nucleotide excision repair (NER), DSB and interstrand crosslink (ICL) repair[34]. Its activity is regulated by the E3 SUMO ligase, SLX4[35]. Importantly, SLX4 SUMOylates the XPF subunit of the Ercc1-XPF complex[36]. The Ercc1-XPF complex also has a role in the repair of Top1 induced damage[37]. Thus, we hypothesized that Ercc1-XPF may also have a role in facilitating DNA repair at heterochromatin after damage induction by ICRF-193, possibly in a SUMO dependent manner.

Interestingly, Ercc1 is efficiently recruited to heterochromatin in cells treated with ICRF-193 (Fig. 7a, b), and this recruitment depends on sumoylation (Fig. S7a). Since the recruitment of Ercc1-XPF can be facilitated by different pathways, we aimed to elucidate the mechanism which regulates this process. For this reason, we used SLX4 Knockout (SLX4KO) MEFs. In the presence of ICRF-193, the ablation of SLX4 abolished the recruitment of Ercc1 to damaged heterochromatin (Fig. 7c). Complementation with SLX4WT led to partial recovery of Ercc1 recruitment at heterochromatin (Fig. 7c), thus corroborating the importance of SLX4 in regulating the recruitment of Ercc1 at heterochromatin upon Top2 inhibition. We further validated these results with transient depletion of SLX4, which prevented the recruitment of Ercc1 to damaged heterochromatin, almost to the same extent as the downregulation of Ercc1 itself (Fig. S7b, c). Interestingly the SLX4-dependent ERCC1 recruitment to heterochromatin upon ICRF-193 does not depend on the SUMO interacting motifs (SIM) of SLX4, as complementation of SLX4 KO MEFs with SLX4ΔSIM also rescues the recruitment of Ercc1 at heterochromatin (Fig. 7c). Therefore, we propose that Ercc1 is recruited to heterochromatin, in an SLX4-dependent manner.

To decipher the potential role of Ercc1 and SLX4 at damaged heterochromatin, we first evaluated whether they have a direct role in DDR. Evidently, depletion of SLX4 or Ercc1 did not affect the levels of γH2AX or Sumo3 at heterochromatin, upon treatment with ICRF-193 (Fig. S7d–f). Nevertheless, given that SLX4 recruits Ercc1, a structure-specific endonuclease, we sought to decipher the role of SLX4 in the formation of DSBs by ICRF-193. To this end, we conducted neutral comet assays. As anticipated, we observed a notable increase in DSBs

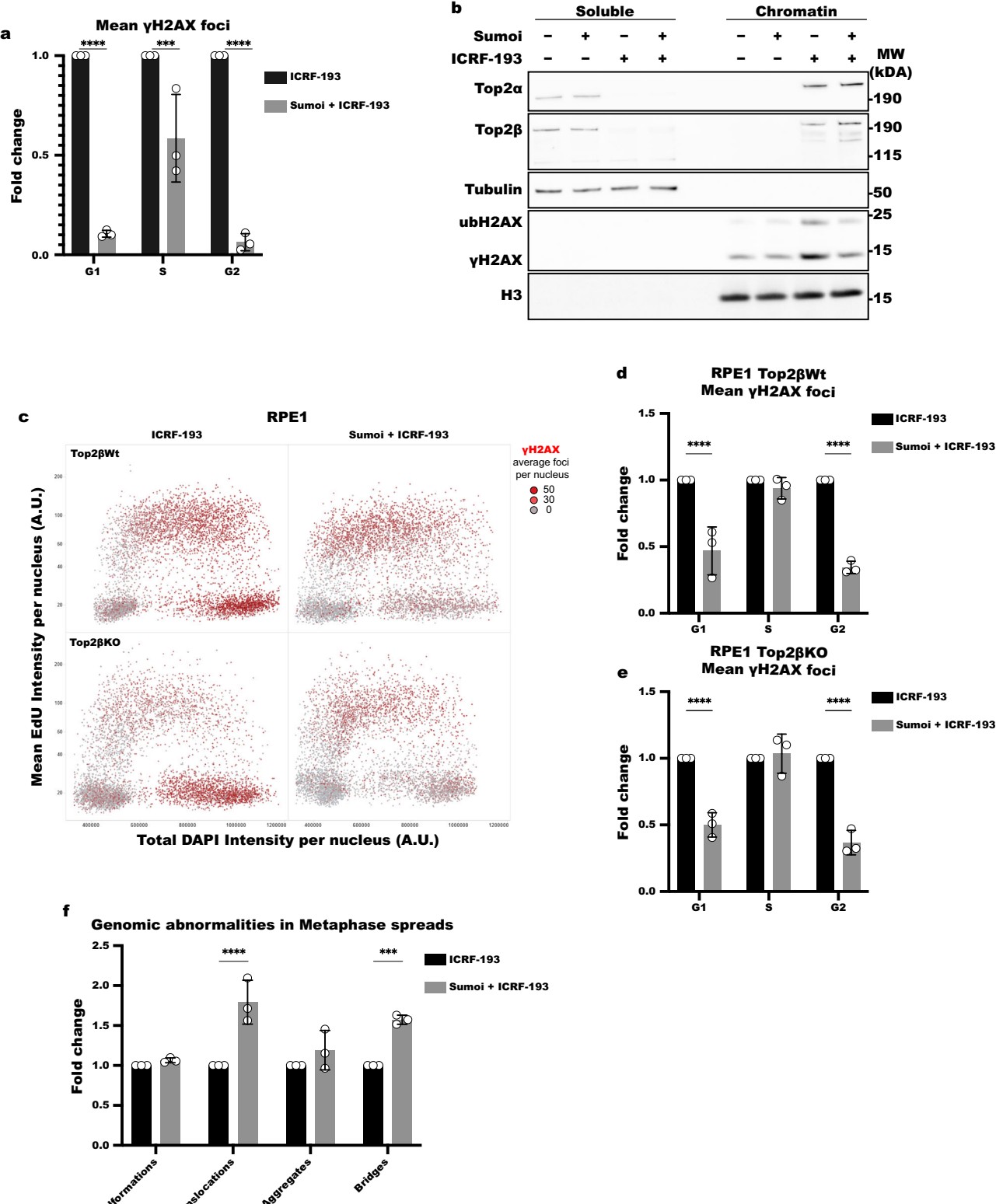

in the wild-type MEFs (Fig. 7d). Conversely, in SLX4-depleted cells (SLX4KO), we did not detect a significant rise in DSB induction in the presence of ICRF-193 (Fig. 7d). Complementation of SLX4KO MEFs with both SLX4wt and SLX4ΔSIM restored DSB induction in agreement with restoration of ERCC1 recruitment (Fig. 7d). These findings indicate that SLX4 plays a role in introducing DSBs into the genome to facilitate repair of damage caused by ICRF-193-mediated trapping of Top2 enzymes around the DNA.

Since the depletion of SLX4 prevents the formation of DSBs, we next assessed whether depletion of Ercc1 or SLX4 affects the genomic stability of cells treated with ICRF-193. To this end, and similarly to Fig. 4a, we quantified the number of malformations, DNA bridges and translocations in the presence of DMSO or ICRF-193, while depleting Ercc1 or SLX4. Indeed, treating NIH3T3 cells with ICRF-193 in the absence of Ercc1 or SLX4, further increased the number of Malformations or translocations, which are indicative of DBSs (Fig. 7e, f).

**Fig. 6 | Sumoylation is necessary for ICRF-193 dependent DDR activation.**
**a** Quantification of the γH2AX foci, in asynchronous NIH3T3 cells treated with DMSO or 5 μM of ICRF-193, with pre-treatment with DMSO or ML-792 sumo inhibitor (Sumoi). **b** Western blot analysis of Top2α and Top2β upon chromatin fractionations of lysates from NIH3T3 cells treated with ICRF-193, with high-salt (500 mM NaCl) containing buffer. Cells were pre-treated with DMSO or Sumoi, prior to addition of ICRF-193. Tubulin & H3 were used as loading controls. **c** QIBC plots of γH2AX in asynchronous Top2βWt and Top2βKO RPE1 cells pre-treated with DMSO or Sumoi and subsequently with ICRF-193. The cell cycle staging was performed based on EdU and DAPI intensity. Each dot represents a single cell, and the color of the dots represents the number of γH2AX foci per cell. **d, e** Quantification of the mean number of γH2AX foci, in asynchronous RPE1 Top2βWt (**d**) or RPE1

Top2βKO (**e**) cells pre-treated with DMSO or Sumoi. **f** Quantification of chromosomal abnormalities, as classified in Fig. 4a, induced by ICRF-193 after pre-treatment with DMSO or Sumoi. Graphs represent average values, and the error bars represent the standard deviation from at least 50 metaphases per condition, and 3 biological replicates. QIBC graphs in this figure represent average values, and the error bars represent the standard deviation from at least 1000 cells per condition, and 3 biological replicates unless stated otherwise. AU arbitrary units. Statistical significance presented by = two-way ANOVA in **a** (*P* values for G1: *p* < 0.0001; S: *p* = 0.0004; G2: *p* < 0.0001), **d** (*P* values for G1: *p* < 0.0001; S: *p* = 0.7629; G2: *p* < 0.0001), **e** (*P* values for G1: *p* < 0.0001; S: *p* = 0.9368; G2: *p* < 0.0001), **f** (*P* values for Malformations: *p* = 0.962; Translocations: *p* < 0.0001; Aggregates: *p* = 0.3355; Bridges: *p* = 0.0003): (*) *p* < 0.05, (**) *p* < 0.01, (***) *p* < 0.001, (****) *p* < 0.0001.

In addition, knockdown of Ercc1 or SLX4 also led to an increase of DNA bridges in the presence of ICRF-193 when compared to Scr (Fig. 7g). Therefore, these results indicate that Ercc1 and SLX4 have a role in preserving ICRF-193-induced genomic stability by introducing chromosome breaks and resolving DNA bridges.

The observed increase of DNA bridges related to heterochromatin in the absence of Ercc1 or SLX4 indicates elevated catenation of pericentromeric heterochromatin among clustered chromosomes. Hence, to unravel if there is an overall increase of inter-connected chromosomes at the level of mitosis, after damage induction by ICRF-193 in the absence of Ercc1 or SLX4, we quantified the number of chromosome clusters between 3 or more chromosomes with inter-chromosomal DNA bridges per metaphase (Fig. 7h, i). In the presence of ICRF-193, the number of clusters significantly increased for both siErcc1 and siSLX4 over the control (Fig. 7j), thus indicating that there are more chromosome clusters with unresolved inter-chromosomal DNA bridges in the absence of these two proteins. Overall, these results reveal a novel role of the Ercc1-XPF complex and SLX4 in the resolution of inter-chromosomal DNA bridges forming between clustered regions of heterochromatin upon Top2 trapping.

In conclusion, our study has uncovered that inhibition of Topoisomerase 2 activity has a multifaced impact on the integrity of heterochromatin and repetitive DNA and sheds light onto the distinct cellular outcomes of Top2 poisons and inhibitors.

## Discussion

Repetitive DNA is often packaged in heterochromatin to maintain its integrity, which is necessary for cell viability and fitness. Centromeric and pericentromeric repeats are essential for proper chromosome segregation, rDNA for growth, and LINEs and other retroelements for the silencing of transposable elements[38–41]. There has been a substantial progress in sequencing and bioinformatic approaches to aid the precise mapping of repetitive elements[41,42]. Nevertheless, it remains unclear whether their repetitive nature, which makes them prone to the formation of secondary structures, their chromatin status, and their propensity to form clusters inside the nucleus leads to elevated topological stress, which can impede DNA replication or chromosome segregation and can jeopardize repeat integrity. Here, we have discovered that repetitive DNA is uniquely sensitive to the inhibition of Topoisomerase 2 activity by the catalytic inhibitor ICRF-193. We demonstrated using microscopy and ChIP-sequencing that ICRF-193 leads to the induction of DNA damage at repetitive regions (Figs. 3b–g & S3b, c, Table S1), and we failed to detect a consistent induction of DNA damage at any other region of the genome (Figs. 3f, g, S3b, c). Thus, we propose ICRF-193 preferentially induces DNA damage at repetitive sequences, without significantly affecting euchromatin.

Top2 poisons, such as etoposide, induce DNA damage throughout the nucleus (Fig. 1a, b), with a preference for CTCF binding regions and active promoters/genes[16,19,21]. This prompts the questions of why is there such a striking difference between poisons and inhibitors, and why does treatment with ICRF-193 lead to DNA damage only in repetitive DNA? One highly likely explanation for the difference between

poisons and inhibitors could be that the DNA damage induced by etoposide and ICRF-193 depends on distinct Top2 isozymes. Although DNA damage induced by etoposide has been shown to depend on both Top2s[21], etoposide induces DNA damage in all stages of the cell cycle and particularly in G1 where primarily Top2β is expressed, but Top2α is not, suggesting that the trapping of Top2β has a major role in the genome instability induced by Top2 poisons. In agreement, DNA breaks induced by etoposide correlate with sites of Top2β binding in the genome[19] and depend on RNA pol II transcription[21]. On the other hand, our data demonstrate that ICRF-193 induced DNA damage depends on Top2α, and perfectly correlates with Top2α expression during the cell cycle. Thus, since the Top2 enzymes are naturally enriched at heterochromatin and repetitive sequences[17,18], a plausible explanation for the specific action of ICRF-193 in these regions is because these genomic regions are either more compacted or tangled and they require elevated levels of Top2 activity to resolve the high topological stress. Therefore, the catalytic inhibition of Top2 activity could result in unresolved torsional stress of pericentric heterochromatin and subsequent occurrence of DNA breaks.

ICRF-193 does not induce DNA damage simply by inhibiting Top2 activity, since the depletion of Top2α rescues the induction of γH2AX foci in the presence of ICRF-193 (Fig. 2a, b, g, h). Therefore, it is more probable that ICRF-193 induces DNA damage by trapping Top2α around the DNA. Nonetheless, the trapping of Top2α on chromatin is necessary but not sufficient to induce the formation of γH2AX foci at heterochromatin, as the ectopic overexpression of Top2α in G1 did not lead to a robust DNA damage induction in the presence of ICRF-193 (Fig. S2l); instead, the Top2 trapping needs to interfere with DNA transactions occurring in S and G2 phases of the cell cycle.

Overall, our results point to the possibility that the mechanism by which ICRF-193 induces DNA damage in S and G2 phases are distinct. In *Xenopus* egg extracts, the trapping of Top2α by ICRF-193 during S phase delayed the activation of replication origin clusters and replication fork progression, while also affecting the spacing of nucleosomes[43,44]. In accordance with these observations, our results in NIH3T3 cells show a decrease in the percentage of early replicating cells and a modest increase in mid and late replicating cells (Fig. S1p). Altogether, these observations indicate that the trapped Top2s by ICRF-193 act as physical barriers, potentially obstructing the progression of the replisome and possibly leading to DNA damage by head on collisions.

The mechanism by which ICRF-193 induces damage preferentially at heterochromatin in G2 cells, could be due to decatenation defects of tangled repeats of the same or different chromosomes which cluster together in the nucleus[45]. The trapping of Top2 enzymes around chromatin in a closed clump can potentially affect the DNA topology leading to chromatin unwinding, thus causing the formation of secondary structures, such as inter-chromosomal full or semi-catenates. In accordance with this hypothesis, previous work demonstrated the stabilization of inter-chromosomal linkages between the rDNA loci of human chromosomes, which also cluster together physiologically, upon Top2 inhibition by ICRF-193[46]. Consequently, this hypothesis

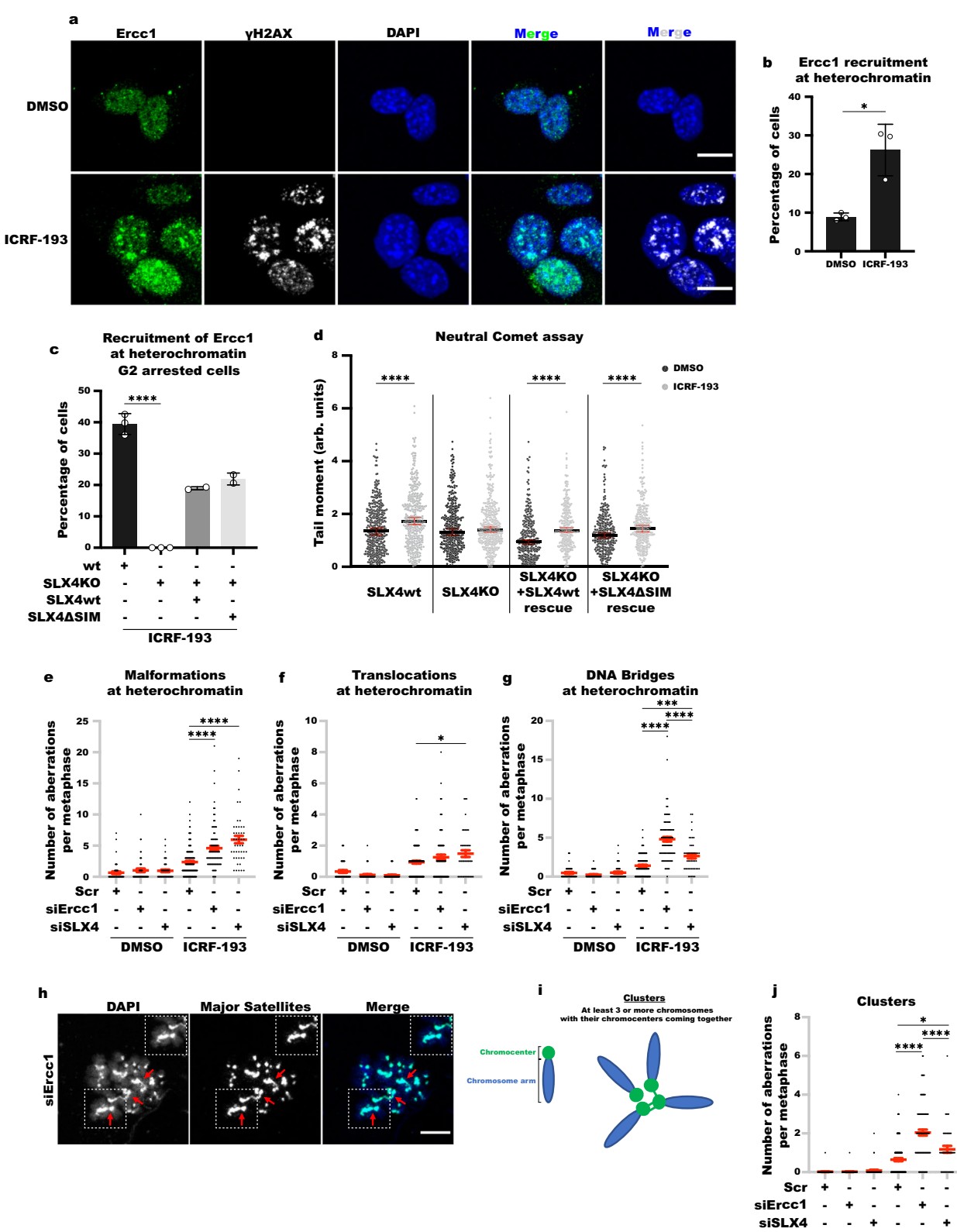

could explain the necessity of structure-specific endonucleases, such as Ercc1, in preserving genomic stability by introducing DSBs and thus preventing the formation of inter-chromosomal DNA bridges and chromosome breaks (Fig. 7g–j).

Another important aspect of this study is the establishment of DNA damage induction by ICRF-193 outside of mitosis. The initial assumptions were that ICRF-193 primarily acts by inhibiting the enzymatic activity of the Top2 enzymes, hence it would damage cells due to

defective decatenation and thus, uneven chromosome segregation to daughter cells[15]. Here, we present an unbiased analysis of γH2AX induction in unperturbed replicating cells based on their cell cycle profile, where we show damage at heterochromatin can occur in interphase, before mitosis begins (Figs. 1d, e, h–j).

We demonstrated that the catalytic activity of Top2α is necessary for the induction of γH2AX (Fig. 5g, h). This observation suggests that DNA damage is induced at the end of the Top2 catalytic cycle when

**Fig. 7 | Ercc1-XPF endonuclease safeguards genomic stability by resolving inter-chromosomal DNA bridges. a** Representative confocal microscopy images of γH2AX (white), Ercc1 (green) and DAPI (blue) in NIH3T3 MEFs, treated with ICRF-193. **b** Quantification of NIH3T3 MEFs with Ercc1 at heterochromatin. **c** Quantification of cells with Ercc1 at heterochromatin in G2-arrested MEFs. For SLX4KO+SLX4wt and SLX4KO + SLX4ΔSIM conditions, mean values are presented from n = 2 biologically independent replicates. At least 100 cells were counted per biological replicate, per condition. **d** Quantifications of tail moment in neutral comet assays performed in MEFs. Bars represent median values from n = 4 independent biological replicates performed for wt and SLX4KO conditions. 100 cells were counted per replicate, per condition. Statistical significance presented by two-sided student t-test between DMSO and ICRF-193 for each independent cell line. Error bars represent 95% CI. *P*-values: SLX4wt: $p = 4.23E-09$; SLX4KO: $p = 0.115$; SLX4KO+SLX4wt rescue: $p = 4.36E-09$; SLX4KO + SLX4ΔSIM rescue: $p = 5.1E-05$. **e**–**g** Quantification of (**e**) malformations, **f** translocations, and (**g**) DNA bridges per metaphase in cells transfected with the indicated siRNAs. The red lines indicate the mean values, and the error bars the standard error of the mean (± SEM).

**h** Representative confocal microscopy images of metaphase chromosome spreads from NIH3T3 cells transfected with siErcc1; Major Satellite repeats (green), DNA (blue); the location of clusters is indicated by the red arrows. **i** Cartoon representing the classification of cluster patterns. **j** Quantification of chromosome clusters per metaphase after transfection with the indicated siRNAs. The red lines indicate the mean values, and the error bars the ±SEM. Graphs represent average values, and the error bars in the graphs represent the standard deviation from at least 50 cells per condition, and 3 independent replicates unless stated otherwise. For confocal images, maximum projection of z-stacks is shown; scale bars = 10 μm. For **e**–**g**, **j**, 1 biological replicate is presented for DMSO. Statistical significance: two-sided student t-test in **b** (*P*-value: $p = 0.0113$); One-way ANOVA in **c** (*P*-value for wt-SLX4KO: $p < 0.0001$), **e** (*P*-values for Scr-siErcc1: $p < 0.001$; Scr-siSLX4: $p < 0.001$; siErcc1-siSLX4: $p = 0.0562$), **f** (*P*-values for Scr-siErcc1: $p = 0.2554$; Scr-siSLX4: $p = 0.0413$; siErcc1-siSLX4: $p = 0.8505$), **g** (*P*-values for Scr-siErcc1: $p < 0.001$; Scr-siSLX4: $p = 0.0006$; siErcc1-siSLX4: $p < 0.001$), and **j** (*P*-values for Scr-siErcc1: $p < 0.001$; Scr-siSLX4: $p = 0.0171$; siErcc1-siSLX4: $p < 0.0001$),: (*) $p < 0.05$, (**) $p < 0.01$, (***) $p < 0.001$, (****) $p < 0.0001$.

both DNA strands are trapped inside the closed Top2 clump. Another possibility is that γH2AX marks the lesions induced by the Top2s themselves which are not promptly ligated due to the subsequent trapping into a clumped form.

The distinction between Topoisomerase poisons and catalytic inhibitors relies predominantly on the ability of a drug to stabilize topoisomerase cleavage complexes. ICRF-193 has long been perceived to act as a catalytic inhibitor of Top2 enzymes, since it does not increase the number of Top2ccs[27]. While accurate, this distinction initially led to the notion that the catalytic inhibitors are less cytotoxic than the Top2 poisons and they simply act by inactivating the Top2 enzymes[47]. Our work provides strong evidence that the absence of Top2 activity leads to a different cellular response than that of the catalytic inhibition of the enzyme. Furthermore, it is evident that there are several similarities between Top2 poisons and catalytic inhibitors. Drugs from both categories have been shown to induce the proteasome dependent degradation of the trapped Top2 enzymes, by utilizing a SUMO/ubiquitin related signaling pathway[31,48,49]. Moreover, the inhibition of sumoylation or ubiquitination leads to a decrease of γH2AX induction in either case[31,33]. Finally, treatment with both types of drugs can stimulate the formation of DSBs and lead to genomic instability[33]. The similarities between the poisons and catalytic inhibitors could suggest that catalytic inhibitors may also act as "non-canonical poisons", even though the mechanism of Top2 inhibition differs.

Given that inhibiting sumoylation results in a significant decrease in γH2AX levels but amplifies genomic instability induced by ICRF-193, we hypothesize that the signaling mechanisms at DNA breaks are affected. This impairment hinders the proper progression of DNA repair processes, thereby contributing to heightened DNA damage and genomic instability. Our findings indicate that the SLX4 SIM domains are not necessary for recruiting ERCC1 or inducing DSBs, implying that SLX4 serves as a scaffold for other SUMO E3 ligases or may overlap with alternative pathways as recently reported[50].

Given that ICRF-193 has such an effect on genomic stability, it could prove to be an effective drug for use in treating cancers. The clinical relevance is highlighted by the fact that ICRF-193 can efficiently induce DNA damage in replicating human cells (Fig. 1h–j). Furthermore, since we observed that cells can commence the G2 phase without first completing DNA replication (Fig. S1q), it is highly probable that they will move into mitosis with defects and therefore carry genomic instability into daughter cells. Similar to mouse cells, our experiments with different human cell lines validated the effects of ICRF-193, including cell cycle dependent induction of DNA damage and the recruitment of DSB repair factors, the absence of Top2ccs stabilization and the importance of the Top2α enzyme in the formation of γH2AX foci. ICRF-193 has yet to be used in the clinical setting for cancer treatment due to its extremely low aqueous solubility, however recent studies have shown that pre-drugs can be used to overcome this obstacle[51,52]. Thus, our findings can be used to analyze and predict the responses of human cells to treatment with ICRF-193. Importantly, the use of Top2 poisons, such as etoposide, in the clinic was shown to be efficient in targeting cancer cells, but unfortunately to also have off-target effects which can lead to secondary malignancies[53]. Consequently, the use of ICRF-193 in the clinical setting could prove to be beneficial over etoposide since its effects are restrained to heterochromatin, suggesting that the risk of DSB formation and subsequent translocations involving potentially oncogenic genes in euchromatin will be dramatically decreased, essentially diminishing the possibility of secondary malignancies such as AML to occur after treatment. In addition, we found that DSBs induced in heterochromatin by ICRF-193 are primarily repaired by the NHEJ repair pathway. These observations are in agreement with previous screens performed on DT40 cells[54]. As a result, treatment with ICRF-193 could have an even greater effect on cancer cells which have lost key molecules of the NHEJ pathway, such as 53BP1 or DNAPK, or other important DNA repair factors presented in this study, such as SLX4, and the Ercc1-XPF complex.

## Methods
### Cell culture
All cell lines, unless stated otherwise, were obtained from the cell banks of Genome Damage and Stability Center (GDSC) or Institut de Génétique et de Biologie Moléculaire et Cellulaire (IGBMC) and cultured in a humidified incubating chamber with 5% $CO_2$ at 37 °C and were passaged before they reached 90% confluency. Human U2OS and mouse embryonic fibroblast (MEFs) cell lines were grown in Dulbecco's Modified Eagle Medium (DMEM, Gibco), supplemented with 10% Fetal Calf Serum (FCS), 100 U/ml Penicillin (Gibco) and 100 U/ml Strepto-mycin (Pen/Strep) and 2 mM L-Glutamine (Gibco). The Ku80 knock-out (Ku80KO) MEFs were a kind gift from Dr. Andre Nussenzweig's lab[55]. Human RPE1-hTERT cells were grown in DMEM/F12 (Gibco) supplemented with 10% FCS, 100 U/ml Pen/Strep, 2 mM L-Glutamine and 125 μg/ml hygromycin. NIH3T3 MEFs were cultured in DMEM supplemented with 10% Newborn Calf Serum (NCS), 100 U/ml Pen/Strep and 2 mM L-Glutamine. HCT116 cells expressing OsTIR1 and Top2α-mAID were a kind gift from Dr. Christian Thomas Friberg Nielsen[14]; they were grown in DMEM, supplemented with 10% FCS, 100 U/ml Pen/Strep, 2 mM L-Glutamine and were selected with 1 μg/ml Puromycin, 7.5 μg/ml blasticidin and 125 μg/ml hygromycin. Selection antibiotics where excluded for experimental cell treatments. Depletion of Top2α was achieved by addition of 500 μM of indole acetic acid (IAA) (Merck; I5148-2G) for 1 h, prior to the addition of ICRF-193. SLX4 WT and KO MEFs were a kind gift from Dr. John Rouse[56]; cells were cultured in DMEM supplemented with 10% Newborn Calf Serum (NCS),

100 U/ml Pen/Strep and 2 mM L-Glutamine. For the generation of stable cell lines, SLX4KO MEFs were transfected with either SLX4wt or SLX4ΔSIM plasmids (Table S4) and 48 h later were FACS sorted for GFP into a pool of cells. SLX4 sorted cell lines were grown and cultured in media supplemented with 2 μg/ml Puromycin.

## Cell treatments

Unless stated otherwise, cells were treated with 5 μM of ICRF-193 (Merck; I4659-1MG) for 4 h. For direct comparison with ICRF-193, cells were treated with 5 μM etoposide (Merck; E1383) for 4 h for the immunofluorescence assay, and with 50 μM for 30' for the Top2cc's isolation assay. For synchronization of cells in G1/S border, double thymidine (Sigma; T1895) block was used, cells were treated with 2 mM thymidine for 18 h, released in drug-free media for 9 h and treated with 2 mM thymidine for 16 h prior to addition of ICRF-193. For synchronization of cells in G2, cells were treated with 10 μM of the CDK1 inhibitor, RO-3306 (Calbiochem; 217699) for 24 h before fixation/collection of cells. The sumoylation inhibitor, 10 μM of ML-70292 (Axon Medchem; 3109), or DMSO as negative control, were added for 90 min prior to addition of ICRF-193. The proteasome inhibitor, 20 μM of MG132 (BIO-TECHNE; 1748), or DMSO as negative control, were added for 1 h prior to addition of ICRF-193. For the inhibition of ATM, 20 μM of Ku-60019 (Selleckchem; S1570), or DMSO were used for 1 h prior to addition of ICRF-193. For the inhibition of ATR, 0.5 μM of VE821 (Calbiochem; 504972), or DMSO were added to cell media overnight prior to addition of ICRF-193. For the simultaneous inhibition of ATM and ATR, VE821 (or DMSO) was first added in cell media overnight, and ATM (or DMSO) was then added 1 h before addition of ICRF-193. For the inhibition of DNAPK, 20 μM of Nu7026 (Merck; N1537-5MG), or DMSO was added to cell media 1 h prior to addition of ICRF-193. For the inhibition of Rad51, 20 μM of B02 (Calbiochem; 553525), or DMSO was added to cell media overnight prior to addition of ICRF-193. For the inhibition of Rad52, 20 μM of (−)-Epigallocatechin (Merck; E3768), or H$_2$O was added to cell media 1 h prior to addition of ICRF-193.

## siRNA transfection by RNAimax

siRNAs (Table S2) where reverse transfected, at a final concentration of 40 nM, with Optimem (Gibco) and Lipofectamine RNAimax reagent (ThermoFisher), according to the manufacturer's instructions. Cell treatments were performed 48 h after the transfection for the down-regulation of Top2α and 72 h after all other siRNAs.

## Plasmid DNA transfections with Lipofectamine 2000

Plasmid DNA (1–2 μg) was transiently transfected with Optimem (Gibco) and Lipofectamine 2000 (ThermoFisher) according to the manufacturer's instructions, for 16 h before treatments.

## Staining for ethynyl-2'-deoxyuridine (EdU) Click-iT

For the detection of replicating cells, EdU (25 μM) was added to the culturing medium 30 min before fixation. Subsequently, Click-iT EdU Alexa Fluor 594 Imaging Kit, or Pacific blue, (Invitrogen) was used, according to manufacturer's instructions. After this step, the protocol for immunofluorescence staining was followed.

## Immunofluorescence

Protocol was followed according to previous publication[45]. In brief, cells were cultured on glass coverslips and fixed with 4% paraformaldehyde (PFA) in 1× PBS for 10 min at room temperature (RT). For the visualization of RPA32, Brca1 and Ercc1 proteins pre-extraction was performed prior to RT fixation, with a 30 second incubation with ice cold 0.1% Triton in 1× PBS and a subsequent 4% PFA in 1× PBS for 10 min on ice. Next, cells were permeabilized in 0.5% Triton in 1× PBS for 10 min, blocked in 5% Bovine Serum Albumin (BSA) for 1 h, then incubated with the primary antibody and secondary antibody (Table S3) in 5% BSA/1× PBS/0.1% Tween), for 1 h each at RT. Finally,

nuclei were counterstained with 1 mg/ml of 4',6-Diamidino-2-Phenylindole, Dihydrochloride (DAPI) for 10 mins.

Images were captured and quantified with a confocal laser scanning microscope (TCS SP8; Leica or LSM 880; Zeiss), using a ×63 objective for at least 50 cells per biological replicate. Brightness & contrast of each color of the confocal microscope images was adjusted accordingly with the ImageJ/Fiji software (version 2.1.0/1.53c).

For QIBC[57], images were captured on a ScanR High-Content Screening Station, which includes an inverted fluorescence Olympus IX83 microscope and a CMOS camera, A ×40 NA 0.6 LUCPLFLN dry objective was used, to acquire at least 1000 cells per biological replicate in identical settings for all samples within the same experiment and non-saturating conditions. Image analysis and generation of representative wide field images was performed using the ScanR software (Version 3.0.1). Nuclei identification and segmentation was performed based on DAPI intensity by an integrated intensity-based object detection module. Background correction was applied and foci segmentation with the use of a spot-detection module. Fluorescence intensities were quantified and portrayed as arbitrary units (A.U.). Cell cycle analysis was based on detection of cell Total DAPI intensity containing 2 C or 4 C DNA (G1 or G2 phase cells respectively) and levels of Mean EdU Intensity (S phase cells). The color-coded scatter plots from asynchronous populations were created with the Spotfire software (version 7.0.1; TIBCO 7.11). A similar number of cells was used to compare sample conditions within the same experiment. Single cells were selected based on object area/Total DAPI intensity or circularity/Total DAPI intensity.

## Fluorescence in situ hybridization for interphase cells and metaphase spreads

**Preparation of metaphase spreads for hybridization with FISH probe.** Protocol was followed according to previous publication[45]. In brief, after 4 h of ICRF-193 treatment, cells were washed once with PBS and colcemid (0.02 μg/ml, 15210040; ThermoFisher) was added to ICRF-193 free growing media for 4 h. The cells were trypsinized and the cell pellet was resuspended in pre-warmed solution of 0.06 M KCl and incubated at 37 °C for 30 mins. Next, the cell pellet was fixed in 3:1 ethanol (EtOH): glacial acetic acid solution, three times and left in the fixative solution overnight at 4 °C. The following day, cell metaphases were spread on ice-cold wet glass slides and left to air dry.

**Preparation of interphase cells for hybridization with FISH probe.** After treatment with ICRF-193 (5 μM) for 4 h, cells were released in drug free culturing medium for 6 h and fixed with 4% PFA. Subsequently, they were washed with PBS, permeabilized with 0.5% Triton.

**Preparation of FISH probe for hybridization.** For the hybridization step, metaphase spreads or interphase cells were treated with RNase A (100 μg/ml in 2× SSC) at 37 °C for 1 h, dehydrated in a series of sequential EtOH washes (70%, 85%, and 100% EtOH) and left to dry. In parallel, the FISH probe was prepared by nick translation from a plasmid containing the major satellite repeats (Table S4). For each coverslip 300 μg of probe DNA and 9 μg of salmon sperm DNA was resuspended in 50% formamide, 2× SSC and 10% dextran sulfate buffer). Finally, the probe and metaphase spreads or cells, were co-denatured for 5 min at 85 °C, and left to hybridize at 37 °C Overnight.

The next day, the samples were washed twice for 20 min in 2× SSC at 42 °C and subsequently incubated with fluorescein anti-biotin antibody (Vector Laboratories, SP-3040; Dilution 1:100) for at least 45 min. Next, they were washed with 2× SSC, incubated with DAPI and mounted with Prolong Gold reagent (Molecular Probes).

Both metaphase spreads and interphase cells were observed and quantified with a confocal laser scanning microscope (LSM 880; Zeiss), using a ×63 objective. For metaphase spreads in Fig. 7e–g, j, in the

DMSO condition at least 50 metaphases counted for Scr, siErcc1 and siSLX4; metaphases counted per condition for the ICRF-193 condition: n = 1 Scr=50, siErcc1 = 22 and siSLX4 = 5; n = 2 Scr=43, siErcc1 = 36 and siSLX4 = 19; n = 3 Scr=35, siErcc1 = 35 and siSLX4 = 21.

### Exo-FISH

Exo-FISH experiments were performed in NIH3T3 MEFs following the protocol published by Saayman et al.[25]. Specifically, 3T3 cells were incubated with 5 μM ICRF-193 for 4 hours and then collected and washed two times in PBS. $2 \times 10^5$ cells per condition were treated with 0.56% KCL for 20 mins and then fixed in methanol-acetic acid (3:1 ratio) for 20 mins. The cells were dropped onto glass slides (60,000 cells/slide) and left to dry overnight at RT, protected from light. The next day, slides were re-hydrated in PBS (5 min) and treated with 0.5 mg/mL RNaseA (10 min, 37 °C), followed by a 5 min wash in PBS. The slides were then treated with 600 mU/μl of Exonuclease III for 45 min, at 37 °C; (Promega M1811), washed in PBS, and run through a sequential ethanol series (70%, 95% and 100%) for 5 min each and left to dry. To detect DNA damage at the chromocenters, the cells were stained with a custom designed probe targeting the Major Satellites (5′-/5Alexa-488N/ AGAAATGTCCACTGTAGGACG-3′, IDT Integrated DNA Technologies) dissolved in hybridization buffer (10 mM Tris-HCl pH 7.2, 0.1 μg/ml Salmon Sperm DNA, 70% formamide and 100 mM Maleic Acid pH 7.5) for 2 h at RT, protected from light. Next, the slides were washed for 15 min in washing buffer 1 (10 mM Tris-HCl pH 7.2, 0.1 % BSA Salmon Sperm DNA, 70% formamide-RT) and washed three times in washing buffer 2 (10 mM Tris-HCl pH 7.2, 150 mM NaCl, 0.08% Tween-20), for 5 min each. Finally, the slides were dehydrated in sequential ethanol series (70%, 95% and 100%) 5 minutes each, left to dry and mounted using Vectashield Antifade Mounting Medium with DAPI (Vecta Laboratories). The Exo-FISH nuclei were imaged with a confocal laser scanning microscope (LSM 880; Zeiss), using a ×63 objective.

### Comet assay

The neutral comet assay protocol was adapted from a previous publication[58]. Briefly, NIH3T3 MEFs were treated with 25 μM ICRF-193 for 4 h, and subsequently washed with PBS and released in drug-free media for 3 h for the release condition. The cells were then collected and washed in ice cold PBS. 50,000 cells were mixed with 1.2% low melting agarose dissolved in PBS and poured onto 1% Agarose/PBS coated glass slides. Cells were lysed overnight at 4 °C in lysis buffer (3.5 M NaCl, 100 mM EDTA, 100 mM Trizma Base, 1% tritonX, 10% DMSO, pH 10). Next, the slides were incubated with neutral electrophoresis buffer (100 mM Tris base, 300 mM Sodium acetate, pH 9). Electrophoresis was then performed at 21 volts (1 volt/cm of the chamber) for 1 h at 4 °C. Next, the slides were immersed in precipitation buffer (1 M $NH_4AC$ in 95% Ethanol) for 30 min, followed by a 5 min incubation in 70% Ethanol. The slides were then left to dry, and the comets were stained with SYBR® Gold. Comet tail moments were scored blinded using the automated Comet Assay IV software (Perceptive Instruments, UK).

### Western blot

**Fractionation of soluble fraction in high salt buffer.** This protocol was followed according to previous publication[59]. Briefly, after treatment with ICRF-193 cells were trypsinized, harvested and then lysed in TEB150 buffer, containing 50 mM HEPES pH 7.4, 500 mM NaCl, 2 mM MgCl2, 5 mM EGTA pH 8, 1 mM dithiothreitol (DTT), 10% glycerol, 0.5% Triton X-100 and Protease Inhibitor Cocktail (PIC; Calbiochem), for 45 min on ice. The chromatin fraction was then pelleted by centrifugation, washed twice in TEB150 buffer and resuspended in 1× Laemmli Sample buffer (Bio-Rad; #1610747), supplemented with 5% β-mercaptoethanol (Merck; M6250). Subsequently, chromatin was briefly incubated at 95 °C and sheared by vortexing.

**Fractionation of nuclear soluble fraction in low salt buffer.** Protocol was followed as in previous publication[60]. In brief, after treatment with ICRF-193, cells were trypsinized and harvested. Cytosolic protein fraction was collected with incubation in hypotonic buffer (10 mM HEPES pH 7, 0.3 M sucrose, 0.5% Triton, 50 nM NaCl and PIC) for 10 min on ice. After centrifugation (1500 × g × 5′), supernatant was collected and the remaining pellet was incubated in the nuclear buffer (10 mM HEPES pH 7, 1 mM EDTA, 0.5% NP-40, 200 mM NaCl and PIC) for 10 min on ice. The supernatant containing the nuclear fraction was collected by centrifugation (13,000 × g × 2′) and the remaining pellet was resuspended in lysis buffer (10 mM HEPES pH 7, 1 mM EDTA, 1% NP-40, 500 mM NaCl, Benzonase [Millipore] and PIC) and sonicated at 10–20% amplitude for three times. The chromatin protein fraction was then collected by centrifugation at 13,000 g × 2 min and collection of the supernatant.

**Western blot analysis.** Equal amounts of protein samples were supplemented with β-mercaptoethanol (Sigma; M6250) and diluted in 1× Laemmli Sample buffer (Bio-Rad; #1610747), incubated at 95 °C for 10 min. Protein samples were run on SDS-Page gels (4–12% NuPAGE Bis-Tris; Invitrogen), in Tris/Glycine/SDS buffer at 80–140 V for 2 h, then transferred on nitrocellulose membrane (0.45 μm; GE Healthcare Lifesciences, 10600062) for 90 min at 400 mA, in ice cold Tris/Glycine buffer, containing 10% methanol. Next, the membranes were blocked for 1 h in 5% milk/PBS-T (Tween-20) for 1 h. Primary antibodies (Table S3) were then incubated on the membranes overnight at 4 °C, in 5% milk/PBS-T. After 3 washes in PBS-T × 5 min each, the membranes were incubated for 1 h in secondary antibody (Table S3) in 5% milk/PBS-T and washed again for 3 times × 5′ each. Protein levels were detected using chemiluminescent Clarity Western ECL reagent (Merck; GERPN2236) and developed using ImageQuant LAS 4000.

### Fluorescence recovery after photobleaching (FRAP)

The FRAP experiments were carried out in a Zeiss LSM880 confocal microscope, containing a FRAP unit and a ×40 1.3NA oil immersion objective lens. NIH3T3 MEFs were seeded in sterile cell culture chambers on coverglass (Sarstedt; 94.6190.402). Top2αWt plasmid was transfected for 16 h, followed by treatment with DMSO or ICRF-193 for 4 h, with 1 h pretreatment with DMSO or Sumo inhibitor (ML−792). During all FRAP measurements, cells were kept at 37 °C in a stage incubator. Protein fluorescence was excited with a laser line at 488 nm, 65.0 intensity and 500 ms interval. The recovery of fluorescence was monitored with the same frequency and exposure time for a total of 100 scans (3 before photobleaching). At least 20 cells were counted per condition, and 3 biological replicates were performed. FRAP kinetics raw data were acquired with a custom-made fiji plug-in macro, created by Dr. Alex Herbert. For the data normalization, three ROI's were created: (1) The photobleached area, which was detected by detection of significant shifts in the intensity of signal using a standard score (The number of standard deviations from the mean). (2) The whole area of nucleus for the detection of loss of fluorescent signal due to photobleaching over time. (3) Background area outside the cell nucleus. FRAP curves were generated by background subtraction, correction of photobleaching overtime and normalization to pre-bleached values. The plot curves of the normalized intensity signal of the photobleached area were created over time after the raw values were analyzed online using the easyFRAP tool with double normalization[61].

Info: (https://easyfrap.vmnet.upatras.gr/easyfrap_web_manual_appendix.pdf)

### Quantitative PCR (qPCR/qRT-PCR) analysis

Isolation of mRNA transcripts was performed using the RNeasy Kit (QIAGEN, Cat No. 74104) according to manufacturer's instructions. Subsequently, RevertAid First Strand cDNA synthesis Kit

(ThermoFisher Scientific, Ref. K1622) was used according to manufacturer's instructions to generate cDNA.

Analysis by qPCR was performed using the Luna Universal qPCR Master Mix (NEB; M3003X) and analyzed on an AriaMx Real-Time PCR instrument (G8830A). The corresponding Ct values were calculated with the Agilent AriaMx software (version 1.71).

## Chromatin immunoprecipitation (ChIP) assay

After treatment with ICRF-193, cells were fixed in 1% PFA for 10 min at RT, followed by the addition of 0.125 M glycine for 10 min, on ice, to stop the cross-linking. Then they were washed twice with pre-cooled 1× PBS and collected by scraping. After centrifugation at $600\,g \times 5'$, they were initially lysed in 50 mM Tris-HCl pH 8, 2 mM EDTA pH 8, 10% glycerol, 0.1% NP-40 and PIC for 5' at 4 °C. Subsequently the sample was centrifuged at $800 \times g \times 5'$, and further lysed in 50 mM Tris-HCl pH 8, 10 mM EDTA pH 8, 1% SDS and PIC. Next, the chromatin was sonicated with E220 Focused-ultrasonicator (Covaris) to create genomic DNA fragments with a size ranging between 300 and 800 bp. 1% of input chromatin was saved and 70 µg of chromatin pre-clearing was done by mixing with pre-blocked Sepharose beads (1:1 vol. mix of Protein A and protein G beads; Invitrogen) for 1 h at 4 °C. Subsequently, 5 µg of antibody were used to immunoprecipitate the chromatin fragments in rotation at 4 °C overnight. The following day, protein A/G Sepharose beads, pre-blocked with BSA, were used to pull down the antibody for 4 h at 4 °C. The beads were washed first twice for 5 min with 20 mM Tris pH8, 2 mM EDTA, 0.5% NP-40, 0.1% SDS, 150 mM NaCl and PIC, then they were washed for 5 min with 20 mM Tris pH8, 2 mM EDTA, 0.5% NP-40, 0.1% SDS, 500 mM NaCl and PIC, and finally for 5 min with 10 mM Tris pH8, 1 mM EDTA, 0.5% NP-40, 0.5% sodium deoxycholate, 250 mM LiCl and PIC at 4 °C. The beads were then eluted twice for 10 min at RT in TE (Tris-HCl 10 mM, EDTA 1 mM), 0.1 M NaHCO₃ and 1% SDS. The eluted DNA samples (and their corresponding 1% Inputs) were then treated with 50 µg/ml of RNase A in the presence of 0.2 M NaCl for 30 min at 37 °C, and followed by treatment with 0.07 mg/ml of Proteinase K. The chromatin fragments were then left to reverse crosslink overnight at 65 °C with shaking and the DNA was then purified with the QIAquick PCR purification kit (QIAGEN; 28104), according to manufactures instructions, and resuspended in elution buffer.

After sample purification, qPCR analysis was performed using the primers from table S5. The corresponding Ct values from samples and inputs were used to calculate the precipitated percentage over input, with the formula $100 \times 2^{Ct(adjusted\,input) - Ct(antibody\,of\,interest)}$. No antibody controls were used to subtract background antibody noise signal from the samples.

## Statistical analysis and figure preparation

Statistical analysis was performed using the GraphPad Prism software. Statistical significance was determined with the respective tests as: (*) $p < 0.05$, (**) $p < 0.01$, (***) $p < 0.001$, (****) $p < 0.0001$; No significant differences are not marked on graphs. For comparison of two groups (Figs. 1b, 2b, d, 3j, 4b, 5b, c, h, 7b, and Supplemental Figs. S1c–e, j, k, n, q, S2b–d, f, g, l, S3k, l, n, o, r, t, S6b, f, S6i, j, k, l, and S7f) unpaired two-tailed t-test was used. For comparison of more than two groups, one-way ANOVA with Dunnett's multiple comparison test, with a single pooled variance was used (Figs. 1c, 2h, i, 3h, 7c, e, f, g, j and Supplemental Figs. S3e, f, g, p), one-way ANOVA with Tukey's multiple comparison test, with a single pooled variance was used (Supplemental Figs. S5c, S6d, g, S7b, c), or two-way ANOVA with Šídák's test, with a single pooled variance was used (Figs. 1i, j, 2f, 6a, d–f, and Supplemental Figs. S1p, S2j, S3i, j). For figure preparation, the Adobe Illustrator and ImageJ (Fiji) were used.

## ChIP-seq analysis

The ChIP-seq libraries were sequenced on an Illumina Hiseq 4000 sequencer as paired-end 100 bases by the GenomEast platform, a member of the 'France Genomique' consortium (ANR-10-INBS-0009). ChIP samples were purified with Agencourt AMPure XP beads (Beckman Coulter) and their concentration was quantified with Qubit (Invitrogen). For the preparation of the ChIP-seq libraries, a total of 10 ng of dsDNA was purified using the MicroPlex Library Preparation kit v3 (C05010001, Diagenode s.a., Seraing, Belgium), according to manufacturer's instructions. RTA and bcl2fastq were performed using base calling and image analysis.

Data analysis was performed in collaboration with the GenomEast platform from IGBMC. Reads were mapped to the mm10 mouse genome assembly using Bowtie2[62] with the parameters: -q -N 1 -X 1000. Since we were interested in repetitive sequences, multiple mapped reads were taken into consideration for the analysis. The peak calling was performed by MACS2 analysis and peak annotation by HOMER, using the default settings. Bigwig files were generated using bamCoverage from deepTools software[63] with RPKM normalization.

**Repeats analysis.** The analysis of repetitive sequences was done according to a previous publication[64]. Reads that were simultaneously mapped for more than one genomic site of the mouse genome, or not mapped at all were aligned to RepBase56 v21.08 (rodent) repetitive sequences. Reads that were mapped uniquely to a single repeat family were annotated with their corresponding family name. The read counts from the two annotation steps per repeat family were added, and the sums of read counts were used to calculate the proportion of each repeat type in every analyzed stage. Two biological replicates were performed, and independent libraries were created, having each library sequenced with at least two technical replicates. Repetitive elements analysis was performed until level 2, showing high reproducibility among the biological replicates. The implemented process in DESeq2 Bioconductor package was used to identify significantly differential repeat sequences. The Wald test P values from the subset of repeats that pass the independent filtering step are adjusted for multiple testing using the procedure of Benjamin and Hochberg[65].

For the bin analysis, the genome was segmented into 10 kb bins and each bin was annotated into active (TSS, TTS, Exons, Introns), repetitive or intergenic sites (based on the sequence content in the center of the bin), using annotatePeaks.pl from Homer software. Differential coverage analysis was performed using DESeq2 (1.38.3) to quantify the number of bins that showed a significant γH2AX enrichment with an over 1.5 log₂ fold increase (cut-off is for ICRF-193 0 h release over DMSO: $\log_2 fc > 1.5$ or $< -1.5$, adjusted p-value < 0.05).

## Top2cc isolation assay

The Top2cc isolation assay was performed using a cc-seq protocol adapted from[16]. In brief, NIH3T3 or U2OS cells were treated with DMSO, ICRF-193 (5 µM for 4 h) or Etoposide (50 µM for 30'). Since they have a slower cell cycle, the U2OS cells were first synchronized in G2 phase with RO-3306 for 20 h prior to addition of DMSO and ICRF-193, or 23.5 h prior to addition of etoposide. Subsequently, the cells were fixed in ice cold 70% EtOH for 10 min, collected by scraping and pelleted by centrifugation. After removing the supernatant, the cells were lysed in 2% SDS, 0.5 M Tris pH 8.0 and 10 mM EDTA buffer at 65 °C for 10 min. After cooling the samples down on ice, Phenol-Chloroform-isoamyl alcohol (25:24:1; Sigma) was supplemented and the samples were mixed with pipetting. The three phases were separated by centrifugation at $2 \times 10^5\,g$ for 20 min and 500 µl of the aqueous phase were transferred in a new tube, and the nucleic acids were precipitated with 1 ml EtOH, washed with 70% ice-cold EtOH and resuspended in TE buffer. Samples were then sonicated to an average of 300–400 bp size fragment with a Covaris sonicator (10% duty cycle, 75 W intensity power incidence, 200 cycles for 20 min). 5% of input was stored and the rest of the samples were then supplemented with a final concentration of 0.3 M NaCl, 0.1% N-Lauroylsarcosine salt and 0.2% Triton-X100 and passed through Miniprep (QIAGEN) silica-fiber membrane

spin columns in order to bind only the Top2cc's and not DNA. Six steps of washes were performed with 10 mM Tris, 0.3 M NaCl and 1 mM EDTA buffer and the Top2cc-bound DNA was eluted in 500 µl of 1 mM EDTA, 0.5% SDS and 10 mM Tris buffer. The samples were incubated at 50 °C for 1 h with 1 mg/ml proteinase K, prior to precipitation overnight at −20 °C with a final concentration of 70% ethanol, 75 mM sodium acetate and 50 µg/ml glycogen. Samples were centrifuged at $2.1 \times 10^4$ g for 1 hour at 60 °C, prior to washing once with 1 ml 70% ethanol, and resuspension in 100 µL TE. Finally, qPCR was performed with primers from Tables S5 and S6 to assess the enrichment of signal in repetitive sequences.

### Reporting summary

Further information on research design is available in the Nature Portfolio Reporting Summary linked to this article.

## Data availability

The raw datasets (ChIP-seq) and processed datasets (ChIP-seq) generated in this study, have been deposited in the Gene Expression Omnibus (GEO) under the accession number GSE256090. [https://www.ncbi.nlm.nih.gov/geo/query/acc.cgi?acc=GSE256090] The repeat analysis data generated in this study are provided in the Supplementary Information file. Source data are provided with this paper.

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

## Acknowledgements

We thank J.M. Dewar, F. Gomez-Herreros, T. Humphrey for critical reading of the manuscript and the members of Soutoglou lab for their constructive comments. Yan Gu and Alex Herbert for their help with FRAP data analysis. Marek Adamowicz for his help with the fractionation assays. Kumar Somyajit for his help with the Spotfire software. A. Nussenzweig for the BRCA1 antibody; Andy Porter for Top2αY805F plasmid; Steve Jackson for the SUMO-RFP plasmids; Alfred Vertegaal & Roman Gonzalez Prieto for SLX4wt and SLX4ΔSIM plasmids; SLX4 WT and KO MEFs were a kind gift from Dr. John Rouse; HCT116 cells expressing OsTIR1 and Top2α-mAID were a kind gift from Dr. Christian Thomas Friberg Nielsen. The E.S. laboratory was supported by the European Research Council (ERC) under the European Union's Horizon 2020 Research and Innovation program (ERC-2015-COG-682939). W.H.G. was supported by the Biotechnology and Biological Sciences Research Council Discovery Fellowship BB/V005081/1. M.J.N. and W.H.G. were supported by The Welcome Trust Investigator Award 200843/Z/16/Z and Welcome Trust Discovery Award 225852/Z/22/Z.

## Author contributions

E.S. and M.A. conceived the study. Most of the experiments and data analysis were done by M.A. Comet assays, exo-FISH and siRad51/siRad52 experiments were done by J.V. FISH experiments for metaphase spreads after Sumoi, siErcc1/siSlx4 and manuscript proofreading were done by K.M. Data analysis for ChIP-seq was done by T.Y. Isolation and analysis of Top2cc was performed by W.H.G. and M.J.N. M.A. and E.S. wrote the manuscript.

## Competing interests

The authors declare no competing interests.
