## [Peer Review File · Nature Communications]

Inhibition of Topoisomerase 2 catalytic activity impacts the integrity of heterochromatin and repetitive DNA and leads to interlinks between clustered repeatsREVIEWER COMMENTS

Reviewer #1 (Remarks to the Author):

In this manuscript by Amoiridis and colleagues, the authors show that catalytic inhibition of Top2 by ICRF-193 genome instability specifically at the heterochromatic and repetitive regions of the genome. Furthermore, they show that the genome instability that is observed in these regions are enhanced upon loss of ERCC1-XPF, the recruitment of which is dependent on SLX4 and SUMO. This is an interesting study which gives mechanistic insights into catalytic inhibition of Top2 results in genome instability and is of wide interest to the field. There are however certain aspects of the manuscript which could benefit from more clarity to make the claims even stronger. Major comments below.

Comments:

1. In figure supplementary figure 1I, the authors claim that they are looking at G2 cells which have not undergone replication. However, there is an increased accumulation of cells in G2 phase after ICRF treatment. I am not sure how cells are accumulating in G2 upon treatment if they have not undergone replication during the treatment of ICRF.
2. In figure S1N, the authors show that there is premature entry of cells in G2 upon ICRF treatment. Could it be possible that that the heterochromatic regions form from premature condensation?
3. It is strange that the timepoints where highest amount of DSBs (4 hrs after release) detected do not co-relate with either BRCA1, 53BP1 and RPA foci dynamics which decreases at this timepoint. The authors suggest that DSBs are detected more robustly only after Top2 removal. End-seq protocol has an extensive de-proteinization step and hence the detection of breaks cannot be dependent of the presence of Top2 on the DNA. Also, background seems to be quite high and the fold change in break intensity doesn't seem to be dramatic as seen by 53BP1 or BRCA1 recruitment.
4. Additionally, I am not sure why in fig3H, the DMSO treated cells have such high level of END-seq peaks.
5. In figure S3 the authors use inhibitors for DNAPK and RAD51 and RAD52 inhibitors to show that NHEJ as opposed to HR or SSA is the preferred pathway to repair damage at heterochromatin induced by ICRF. The authors should solidify the data using genetic ablation of these factors.
6. The authors show in fig6 that H2AX foci is dependent of SUMOylation. However, the chromosomal aberrations go up upon SUMO_i treatments suggesting that the breaks are not repaired. Is it the signaling mechanisms that is getting affected? How do the authors envision this happening?
7. The authors further hypothesize that the SUMO ligase activity of SLX4 could be important to ERCC1-XPF recruitment but show no data supporting this. Does the SUMO mutant of SLX4 prevent ERCC1-XPF1 recruitment? Does it also affect SUMOylation and H2AX formation upon ICRF treatment and release? Furthermore the authors should also perform experiments to assess if there are more breaks in these conditions at heterochromatic and repetitive regions by End-seq.

Reviewer #2 (Remarks to the Author):

This comprehensive and detailed study reports the differential effects of type II topoisomerase inhibition at specific chromatin compartments, suggesting the resolution of trapped Top2 and unresolved catenates at heterochromatin and repetitive sequences is catalyzed by the XPF endonuclease and requires SUMOylation. Beyond probing into the mechanistic consequences of Top2 trapping, these observations also suggest a role of Top2 in resolving inter-chromosomal catenates at repetitive elements.

The paper tackles an important biological question from many angles and provides mechanistic details of the cellular response to topoisomerase inhibitions, which are commonly used in the clinic. Although the paper is quite long, it is clearly written, the experiments are rationally linked

and the reported experiments utilize appropriate state-of-the-art methodology. Most of the conclusions are strongly supported by the data. However, there is a concern related to the designation of the genomic regions enriched in DNA damage. The notion that DNA damage is enhanced primarily in repetitive sequences residing in heterochromatin can be better clarified and supported, as described below. The paper can be also improved by addressing several other issues, mostly minor.

Comments and suggestions:

1. The paper provides several lines of evidence supporting the conclusion that DNA damage by Top2 inhibitors primarily affects heterochromatin repetitive sequences, including colocalization of gamma H2AX signal with chromocenters, enhanced gamma H2AX signals in repeated sequences and localization of DNA breaks using END-Seq. However, each of these approaches has limitations. For example, the imaging-based studies utilize colocalization to chromo-centers as a surrogate to heterochromatin, but the association between chromo-centers and heterochromatin was not determined directly in cells treated with Top2 inhibitors. This point could be further supported by using heterochromatin epigenetic marks (e.g. methylated histones, heterochromatin-associated proteins) in addition to staining for chromo-centers.
2. The sequencing analyses reported in the paper suggest that both gamma H2AX and END-Seq reads were increased in some categories of repetitive sequences. However, data are only presented for some categories of repeated sequences, and it is unclear if only those categories are affected. If only some categories were affected by ICRF193 whereas others were not, this conclusion should be stated.
3. The paper does not present direct measurements that address the selectivity of the response to ICRF193. The paper's conclusions would be strongly corroborated by including a direct quantification of genome-wide signals in non-repetitive genomic regions in addition to the quantification of such signals in repetitive sequences. For example, in Figure 3B, it is unclear if the observed increased signal in the repetitive sequences implies that gamma-H2AX was specifically recruited to repetitive sequences, or just exhibits an increased signal as a consequence of a genome-wide enhancement of gamma-H2AX binding. A more detailed analysis that clearly indicates how copy number was taken into account would improve the paper.
4. Related to the above, it would be useful to include a section describing how data aligned to mm9 were analyzed to compensate for the documented gaps and inaccuracies in this genome build, especially when repetitive sequences were interrogated.
5. The paper reports some interesting cell cycle effects of top2 inhibitors, but the flow analyses shown in the paper do not unequivocally support the conclusion that replication of heterochromatin slowed in response to exposure to ICRF193. Such a conclusion necessitates either a direct measurement of DNA synthesis rates or, at a minimum, a time course of flow-based pulse-chase analyses. This conclusion should be further substantiated or removed.

Minor

Figure 1B: check the apparently low statistical significance of the global vs heterochromatin damage in etoposide. It seems that these values strongly differ from each other.

Reviewer 1:

This is an interesting study which gives mechanistic insights into catalytic inhibition of Top2 results in genome instability and is of wide interest to the field.

We thank the reviewer for finding our study of wide interest.

1. In figure supplementary figure 1L, the authors claim that they are looking at G2 cells which have not undergone replication. However, there is an increased accumulation of cells in G2 phase after ICRF treatment. I am not sure how cells are accumulating in G2 upon treatment if they have not undergone replication during the treatment of ICRF.

We understand the point of the reviewer. Indeed, treatment with ICRF-193 leads to accumulation of cells in G2. The accumulation of cells in G2 is expected, due to the Top2 α -dependent arrest associated with faithful sister chromatid segregation¹. Nevertheless, the experiment shown in supplementary figure 1L, was designed to test whether passing from S phase is absolutely required for γ H2AX induction by ICRF-193 treatment. For this reason, by adding the EdU before treatment with ICRF-193, we could ensure we analysed the population that arrested in G2, in which replication had already finished before ICRF-193 was added (the EdU- G2 cells; selection of 4N cells based on DAPI content) (new Fig. S1l). Interestingly, ICRF-193 induces γ H2AX foci formation in 92.6% of the EdU- G2 population ($p=4.39 \times 10^{-7}$), revealing that active replication is not necessary during the treatment for DNA damage to be induced, and that the DNA damage is not necessarily passed on to G2 cells from the previous S phase (new Fig. S1m and S1n).

This does not mean that ICRF-193 does not induce DNA damage while chromocenters are replicating. As we have described in the paper and expanded upon in the discussion, we believe that ICRF-193 acts in two steps to induce damage in cells. The first step occurs during S phase and is possibly due to replication collisions, whereas the second step is post-replicative and likely happens due to post-replicative decatenation problems.

We have now modified the text to make these points clearer in the results section, in lines 161-164.

2. In figure S1Q, the authors show that there is premature entry of cells in G2 upon ICRF treatment. Could it be possible that that the heterochromatic regions form from premature condensation?

In figure S4, we show that the overall pH3S10 levels stay low during treatment with ICRF-193 and only increase after the release of the drug. Therefore, we do not believe that the chromocenters are prematurely condensed because otherwise, due to their high compaction levels, we would be able to detect high levels of pH3S10 by Western blot. In accordance with this, the levels of H3K9me3 and HP1 α at heterochromatin do

not change upon ICRF-193 treatment, (Fig. S1b-S1d). We now describe these new data in lines 114-117.

3. It is strange that the timepoints where highest amount of DSBs (4 hrs after release) detected do not co-relate with either BRCA1, 53BP1 and RPA foci dynamics which decreases at this timepoint. The authors suggest that DSBs are detected more robustly only after Top2 removal. End-seq protocol has an extensive deproteinization step and hence the detection of breaks cannot be dependent of the presence of Top2 on the DNA. Also, background seems to be quite high and the fold change in break intensity doesn't seem to be dramatic as seen by 53BP1 or BRCA1 recruitment.

4. Additionally, I am not sure why in fig3H, the DMSO treated cells have such high level of END-seq peaks.

We agree with the reviewer that the dynamics of 53BP1, RPA32 and BRCA1 do not fit entirely with the results from END-seq. What is clear is that both the techniques show the presence of DSBs. In addition, the genomic instability (particularly the chromosomal translocations) observed in metaphase spreads, 4h after we have removed ICRF-193, also indicates the formation of DBSs.

However, because the mapping of DSBs in repetitive sequences after an extensive deproteinization step used in END-seq can be rather challenging and the results showing high levels of background at pericentric heterochromatin are difficult to interpret, we replaced End-seq with two alternative techniques. First, we performed neutral comet assays, a method which specifically detects the enrichment of DSBs in the nucleus. The results agree with our END-seq findings, showing an enrichment of DSBs both upon treatment with ICRF-193 and also after the release of the drug (Fig. 3h). To supplement this observation, we also performed Exo-FISH, a recently developed imaging-based technique enabling the visualization of DSBs in repetitive sequences within the nucleus^{2,3}. This verified that the breaks induced by ICRF-193 are located at the pericentric heterochromatin of the chromocenters (Fig. 3i & 3j).

We have now added these results in lines 308-317.

5. In figure S3 the authors use inhibitors for DNAPK and RAD51 and RAD52 inhibitors to show that NHEJ as opposed to HR or SSA is the preferred pathway to repair damage at heterochromatin induced by ICRF. The authors should solidify the data using genetic ablation of these factors.

Following the reviewer's recommendations, we performed siRNAs-dependent depletions of DNAPK, RAD51 and RAD52 (included in New Fig. S3). As expected, depletion of DNAPK led to a delayed γ H2AX resolution compared to siScramble (Scr) 4h after the release from ICRF-193 (Fig. S3p). Depletion of RAD52 did not have any significant effect, verifying the results from the RAD52 inhibition (Fig. S3p, S3s). Interestingly, depletion of RAD51 led to a delay in γ H2AX kinetics (Fig. S3p), which we did not previously detect by Rad51 inhibition (Fig. S3t), possibly because as shown

in other studies⁴, the B02 inhibitor reduces RAD51 foci formation but does not entirely abolish them. Nevertheless, genetic ablation of Ku80 showed overall the most prominent effect in delaying DNA repair, exemplified by the γ H2AX signal after 6h release of ICRF-193 (Fig. S3n). These data agree with previous screens performed on DT40 cells that show a major role of NHEJ factors and a lesser importance of HR factors in the sensitivity to ICRF-193⁵. However, since we do detect a role of Rad51 in the kinetics of γ H2AX, we have now toned down our previous statement in lines 331-342 & 724-725.

6. The authors show in fig6 that H2AX foci is dependent of SUMOylation. However, the chromosomal aberrations go up upon SUMOi treatments suggesting that the breaks are not repaired. Is it the signaling mechanisms that is getting affected? How do the authors envision this happening?

We understand the point of the reviewer. However, we are not surprised by this finding because various studies, including those conducted in Steve Jackson's lab, demonstrate the dependence of DNA damage response (DDR) on sumoylation⁶. Considering that inhibiting sumoylation results in a significant decrease in γ H2AX levels but amplifies genomic instability induced by ICRF-193, we hypothesize that the signalling mechanisms at DNA breaks are affected. This impairment hinders the proper progression of DNA repair processes, thereby contributing to heightened DNA damage and genomic instability. We now explain this better in the text in lines 694-698.

7. The authors further hypothesize that the SUMO ligase activity of SLX4 could be important to ERCC1-XPF recruitment but show no data supporting this. Does the SUMO mutant of SLX4 prevent ERCC1-XPF1 recruitment? Does it also affect SUMOylation and H2AX formation upon ICRF treatment and release? Furthermore, the authors should also perform experiments to assess if there are more breaks in these conditions at heterochromatic and repetitive regions by End-seq.

We thank the reviewer for the comment. To address this point, we used SLX4 knock-out (SLX4KO) MEFs rescued with SLX4wt to first validate that Ercc1 recruitment depends on SLX4 (Fig.7c). We also performed neutral comet assay and observed that the formation of DSBs is hindered, upon depletion of SLX4 (Fig. 7d). This result shows the importance of SLX4 in recruiting structure-specific endonuclease, such as Ercc1-XPF, to induce DSBs and facilitate DNA repair to prevent the formation of DNA bridges in catenated repetitive regions of heterochromatin.

To assess how SLX4 or Ercc1 affect the DDR at heterochromatin, after treatment with ICRF-193, we quantified the levels of γ H2AX or SUMO3 at heterochromatin upon their downregulation. Overall, depletion of Ercc1 or SLX4 did not affect the levels of γ H2AX or SUMO-3 at heterochromatin (Fig. S7d-S7f), showing that it is not the E3 SUMO-ligase activity of SLX4 that regulates the DDR after Top2 inhibition. In

accordance with these observations, the rescue of SLX4KO cells with SLX4 which lacks the SUMO interacting motifs (SLX4 Δ SIM) still leads to the recruitment of Ercc1 at heterochromatin (Fig. 7c) and an increase of DSBs (Fig. 7d). This observation shows that the SUMO ligase activity of SLX4 is not responsible for the recruitment of Ercc1 *per se* and it is in line with the fact that the interaction domain of SLX4 with ERCC1-XPF is on the N-terminus of the protein, far from the SIMs. Consequently, our findings indicate that the SLX4 SIM domains are not necessary for recruiting ERCC1 or inducing DSBs, implying that SLX4 serves as a scaffold for other SUMO E3 ligases or may overlap with alternative pathways as recently reported⁷. We now describe the new data in lines 528-554 discuss them in lines 656-659, 698-701.

Reviewer 2:

The paper tackles an important biological question from many angles and provides mechanistic details of the cellular response to topoisomerase inhibitions, which are commonly used in the clinic.

We are pleased the reviewer finds the focus of our study to be of importance and comprehensive.

Comments and suggestions:

1. The paper provides several lines of evidence supporting the conclusion that DNA damage by Top2 inhibitors primarily affects heterochromatin repetitive sequences, including colocalization of gamma H2AX signal with chromocenters, enhanced gamma H2AX signals in repeated sequences and localization of DNA breaks using END-Seq. However, each of these approaches has limitations. For example, the imaging-based studies utilize colocalization to chromo-centers as a surrogate to heterochromatin, but the association between chromo-centers and heterochromatin was not determined directly in cells treated with Top2 inhibitors. This point could be further supported by using heterochromatin epigenetic marks (e.g. methylated histones, heterochromatin-associated proteins) in addition to staining for chromocenters.

We thank the reviewer for the comment. We now show that the percentage of cells where H3K9me3 and HP1 α colocalize with the chromocenters is not significantly altered upon treatment with ICRF-193 (Fig. S1b-S1d). Thus, these findings further corroborate our observations of Top2 inhibition by ICRF-193 in heterochromatin. We describe these data in lines 112-117.

2. The sequencing analyses reported in the paper suggest that both gamma H2AX and END-Seq reads were increased in some categories of repetitive sequences. However, data are only presented for some categories of repeated sequences, and it is unclear if only those categories are affected. If only some categories were affected by ICRF193 whereas others were not, this conclusion should be stated.

We understand the concern of the reviewer. We apologise for the misunderstanding. The number of classes of repeats in the genome is vast (see Table S1). We see a robust increase particularly in the major and minor satellites, and rDNA repeats. We also see an enrichment in distinct, but not all, LINE or SINE elements. For this reason, we have toned down our previous statement, and now clearly state it in our main text mentioned in lines 284-288.

3. The paper does not present direct measurements that address the selectivity of the response to ICRF193. The paper's conclusions would be strongly corroborated by including a direct quantification of genome-wide signals in non-repetitive genomic regions in addition to the quantification of such signals in repetitive sequences. For example, in Figure 3B, it is unclear if the observed increased signal in the repetitive sequences implies that gamma-H2AX was specifically recruited to repetitive sequences, or just exhibits an increased signal as a consequence of a genome-wide enhancement of gamma-H2AX binding. A more detailed analysis that clearly indicates how copy number was taken into account would improve the paper.

We understand the point of the reviewer. To exclude the possibility that the increase in γ H2AX we see is a consequence of a general increase in γ H2AX signal throughout the genome after ICRF-193, and to further verify the specific effects in heterochromatin vs euchromatin we segmented the genome into 10kb bins and annotated them into active (TSS, TTS, Exons, Introns), repetitive or intergenic sites (based on the sequence content in the centre of the bin), using `annotatePeaks.pl` from Homer software. To understand if ICRF-193 induces damage preferentially in repetitive sequences over active sites, we performed a differential coverage analysis using DESeq2 (1.38.3) to quantify the number of bins that showed a significant γ H2AX enrichment with an over 1.5 \log_2 fold increase (cut-off is for ICRF-193 0h release over DMSO: $\log_2fc > 1.5$ or < -1.5 , adjusted p-value < 0.05) (Fig. S3a). This comparison showed ICRF-193 does not lead to a genome wide enrichment of γ H2AX and that γ H2AX was significantly enriched only in a fraction of the bins (Fig. S3a). To investigate where in the genome the majority of bins are annotated, we visualized the overall distribution of the three main categories of annotated bins in the genome, which shows that they are equally distributed (Fig. 3Sb). Importantly, the majority of the bins with a significant induction of γ H2AX, upon treatment with ICRF-193, are repetitive sites, with only a small portion of bins annotated to active sites (Fig. S3c). These data are now included in lines 276-284.

Since we find only a small portion of the damaged regions to be localized in active regions, we believe it is safe to assume that the effects of ICRF-193 induce γ H2AX preferentially in repetitive regions. These observations are in agreement with our data from IF analysis (Fig 1) and our new data using Exo-FISH (updated Fig. 3i-3j) in our revised manuscript.

4. Related to the above, it would be useful to include a section describing how data aligned to mm9 were analyzed to compensate for the documented gaps and inaccuracies in this genome build, especially when repetitive sequences were interrogated.

To avoid this issue, we repeated our analysis using mm10 genome, and now use this analysis for our revised manuscript. The mm10 build gave us similar results to mm9 (our original submission). The previously presented graphs (Fig. 3a-3g) and the 10 kb segmentation data (updated Fig. S3a-S3c) are generated from analysis using the mm10 assembly. Importantly, we always compare the read enrichment of ICRF-193 over DMSO with the same assembly, so even if we miss information with the gaps, we believe it is safe to assume that we do not create any false positive results with our interpretations.

5. The paper reports some interesting cell cycle effects of top2 inhibitors, but the flow analyses shown in the paper do not unequivocally support the conclusion that replication of heterochromatin slowed in response to exposure to ICRF193. Such a conclusion necessitates either a direct measurement of DNA synthesis rates or, at a minimum, a time course of flow-based pulse-chase analyses. This conclusion should be further substantiated or removed.

We agree with the reviewer. We did not elaborate much on this matter since previous publications have shown something similar^{8,9} which is in accordance with our data. As suggested, we have now removed the statement from our text.

Minor

Figure 1B: check the apparently low statistical significance of the global vs heterochromatin damage in etoposide. It seems that these values strongly differ from each other.

We thank the reviewer for bringing this to our attention. Following the reviewer's recommendation, we have repeated our statistical analysis (student *t*-test) in Fig. 1b and realised the significance stars were mixed between ICRF-193 and etoposide conditions. We have now corrected them in the graph.

References:

- 1 Deiss, K. *et al.* A genome-wide RNAi screen identifies the SMC5/6 complex as a non-redundant regulator of a Topo2a-dependent G2 arrest. *Nucleic Acids Res* **47**, 2906-2921, doi:10.1093/nar/gky1295 (2019).
- 2 Saayman, X., Graham, E., Leung, C. W. B. & Esashi, F. exo-FISH: Protocol for detecting DNA breaks in repetitive regions of mammalian genomes. *STAR Protoc* **4**, 102487, doi:10.1016/j.xpro.2023.102487 (2023).

- 3 Saayman, X., Graham, E., Nathan, W. J., Nussenzweig, A. & Esashi, F. Centromeres as universal hotspots of DNA breakage, driving RAD51-mediated recombination during quiescence. *Mol Cell* **83**, 523-538 e527, doi:10.1016/j.molcel.2023.01.004 (2023).
- 4 Shkundina, I. S., Gall, A. A., Dick, A., Cocklin, S. & Mazin, A. V. New RAD51 Inhibitors to Target Homologous Recombination in Human Cells. *Genes (Basel)* **12**, doi:10.3390/genes12060920 (2021).
- 5 Maede, Y. *et al.* Differential and common DNA repair pathways for topoisomerase I- and II-targeted drugs in a genetic DT40 repair cell screen panel. *Mol Cancer Ther* **13**, 214-220, doi:10.1158/1535-7163.MCT-13-0551 (2014).
- 6 Jackson, S. P. & Durocher, D. Regulation of DNA damage responses by ubiquitin and SUMO. *Mol Cell* **49**, 795-807, doi:10.1016/j.molcel.2013.01.017 (2013).
- 7 Hertz, E. P. T. *et al.* The SUMO-NIP45 pathway processes toxic DNA catenanes to prevent mitotic failure. *Nat Struct Mol Biol* **30**, 1303-1313, doi:10.1038/s41594-023-01045-0 (2023).
- 8 Gaggioli, V., Le Viet, B., Germe, T. & Hyrien, O. DNA topoisomerase IIalpha controls replication origin cluster licensing and firing time in *Xenopus* egg extracts. *Nucleic Acids Res* **41**, 7313-7331, doi:10.1093/nar/gkt494 (2013).
- 9 Germe, T. & Hyrien, O. Topoisomerase II-DNA complexes trapped by ICRF-193 perturb chromatin structure. *EMBO Rep* **6**, 729-735, doi:10.1038/sj.embor.7400465 (2005).

REVIEWERS' COMMENTS

Reviewer #1 (Remarks to the Author):

The authors have satisfactorily addressed all my concerns and I recommend this manuscript for publication.

Reviewer #2 (Remarks to the Author):

The revision has addressed my concerns. Below are some minor suggestions:

Line 3 correlates

Line 305 did you mean "in a cell cycle-dependent manner"?

Line did you mean "has a minor role in the induction of DNA damage"?

Line 638 Xenopus

Point by point response to the reviewers:

Reviewer #1 (Remarks to the Author):

The authors have satisfactorily addressed all my concerns and I recommend this manuscript for publication.

We thank the reviewer for the recommendation to publish the manuscript.

Reviewer #2 (Remarks to the Author):

The revision has addressed my concerns.
Below are some minor suggestions:

Line 3 correlates

This is now corrected

Line 305 did you mean "in a cell cycle-dependent manner"?

This is now accordingly changed

Line did you mean "has a minor role in the induction of DNA damage"?

This is now accordingly changed

Line 638 Xenopus

This word has been corrected.